# Assimilation of surface observations in a transient marine ice sheet model using an ensemble Kalman filter

Fabien Gillet-Chaulet[1]

[1]Univ. Grenoble Alpes, CNRS, IRD, IGE, F-38000 Grenoble, France

**Correspondence:** Gillet-Chaulet Fabien (fabien.gillet-Chaulet@univ-grenoble-alpes.fr)

**Abstract.**

Marine based sectors of the Antarctic Ice Sheet are increasingly contributing to sea level rise. The basal conditions exert an important control on the ice dynamics and can be propitious to instabilities in the grounding line position. Because the force balance is non-inertial, most ice flow models are now equipped with time-independent inverse methods to constrain the basal conditions from observed surface velocities. However, transient simulations starting from this initial state usually suffer from inconsistencies and are not able to reproduce observed trends. Here, using a synthetic flow line experiment, we assess the performance of an ensemble Kalman filter for the assimilation of transient observations of surface elevation and velocities in a marine ice sheet model. The model solves the shallow shelf equation for the force balance and the continuity equation for ice thickness evolution. The position of the grounding line is determined by the floatation criterion. The filter analysis estimates both the state of the model, represented by the surface elevation, and the basal conditions, with the simultaneous inversion of the basal friction and topography. The idealized experiment reproduces a marine ice sheet that is in the early stage of an unstable retreat. Using observation frequencies and uncertainties consistent with current observing systems, we find that the filter allows to accurately recover both the basal friction and topography after few assimilation cycles with relatively small ensemble sizes. In addition it is found that assimilating the surface observations has a positive impact to constrain the evolution of the grounding line during the assimilation window. Using the initialised state to perform century-scale forecast simulations, we show that grounding line retreat rates are in agreement with the reference, however remaining uncertainties in the basal conditions may lead to significant delays in the initiation of the unstable retreat. These results are encouraging for the application to real glacial systems.

## 1 Introduction

Despite recent significant improvements in ice-sheet models, the projected magnitude and rate of the Antarctic and Greenland ice sheets contribution to 21st century sea-level rise (SLR) remains poorly constrained (Church et al., 2013). Improving our

ability to model the century-scale magnitude and rates of mass loss from marine ice sheets remains a key scientific objective (Scambos et al., 2017).

Improving SLR estimates requires, amongst others, to correctly model the dynamics of the grounding line (GL), i.e. the location where the ice detaches from its underlying bed and goes afloat on the ocean (Durand and Pattyn, 2015). In the GL vicinity, the stress regime changes from a regime dominated by vertical shearing in the grounded part to a buoyancy driven flow dominated by longitudinal stretching and lateral shearing (Pattyn et al., 2006; Schoof, 2007). Because this transition occurs on horizontal dimensions that are smaller than the typical grid size of large scale ice sheet models, many studies have focussed on the ability of the numerical model to properly simulate grounding line migration using synthetic experiments (e.g., Vieli and Payne, 2005; Durand et al., 2009; Gladstone et al., 2012; Seroussi et al., 2014). Two Marine Ice-Sheet Model Intercomparison Projects (MISMIP) have allowed to identify the minimum requirements to properly resolve GL motion: *(i)* inclusion of membrane stresses and *(ii)* a sufficiently small grid size or a subgrid interpolation of the GL (Pattyn et al., 2012, 2013). These results suggest that, in realistic applications, the numerical error could be reduced below the errors associated with uncertainties in the model initial state, in the model parameters and in the forcings from the atmosphere and ocean.

For obvious reasons of inaccessibility, the basal conditions (topography and friction) are an important source of uncertainties. Because of the intrinsic instability of marine ice sheets resting over a seaward up-sloping bed, the resolution of the bed topography in the coastal regions can significantly affect short term ice-sheet forecasts (Durand et al., 2011; Enderlin et al., 2013). Analytical developments have shown that the flux at the grounding line depends on the friction law and its coefficients (Schoof, 2007; Tsai et al., 2015). The sensitivity of model projections to the basal friction has been confirmed by several numerical studies both on synthetic and real applications (Joughin et al., 2010; Ritz et al., 2015; Brondex et al., 2017, 2018). In particular, Brondex et al. (2017) have shown that, for unbuttressed ice sheets, spatially varying friction coefficients can lead to stable GL positions also in up-sloping bed regions.

Uncertainties in the model state and parameters can be reduced by data assimilation (DA). The objective of formal DA methods is to update the model using observations in a framework consistent with the model, the data and their associated uncertainties (Bannister, 2017). Most ice flow models are now equipped with variational methods to constrain the basal conditions from surface observations (e.g. MacAyeal, 1993; Vieli and Payne, 2003; Larour et al., 2012; Gillet-Chaulet et al., 2012). However most studies perform "snapshot" calibrations, where the inversion is performed at a unique initial time step. The state of the model produced from this calibration is therefore sensitive to inconsistencies between the different datasets. The resulting transient artefacts are usually dissipated during a relaxation period where the model drift from the observations.

Because historic remote sensing data collections are spatially incomplete as well as temporarily sparse, most distributed maps are mosaicked, stacked or averaged to maximize the spatial coverage at the expense of the temporal information (Mouginot et al., 2012). However, in the last few years, the development of spaceborne ice-sheet observations has entered a new era with the launch of new satellite missions, considerably increasing the spatial and temporal resolution of surface observations. Because they require linearized versions of the forecast model and of the observation operator, extending the existing variational methods implies important numerical developments (*e.g.* Goldberg et al., 2016; Larour et al., 2016; Hascoët and Morlighem, 2018). In Goldberg and Heimbach (2013), a time-dependent adjoint ice flow model is derived using a source-to-

source algorithmic differentiation software combined with analytical methods. The DA capabilities are illustrated with a suite of synthetic experiments, including the simultaneous inversion of the basal topography and friction from surface observations and the assimilation of transient surface elevations to retrieve initial ice thicknesses. In a real-world application to a region of West Antarctica, they show that assimilating annually resolved observations of surface height and velocities between 2002 and 2011 allows to improve the model initial state, giving better confidences in projected committed mass losses (Goldberg et al., 2015). Because of the complexity of the code, Larour et al. (2014) use an operator-overloading approach to generate the adjoint and assimilate surface altimetry observations from 2003 to 2009 to constrain the temporal evolution of the basal friction and surface mass balance of the Northeast Greenland Ice Stream.

Ensemble DA methods, based on the ensemble Kalman filter (EnKF), have been successful in solving DA problems with large and non-linear geophysical models. Comparative discussions of the performances and advantages of variational and ensemble DA methods can be found in, *e.g.* Kalnay et al. (2007), Bannister (2017) and Carrassi et al. (2018). As they aim at solving similar problems, a recent tendency is to combine both methods to benefit from their respective advantages.

EnKF approximates the state and the error covariance matrix of a system using an ensemble that is propagated forward in time with the model, avoiding the computation of the covariance matrices and the use of linearised or adjoint models. Contrary to time-dependent variational methods where the objective is to find the model trajectory that minimizes the difference with all the observations within an assimilation window, EnKF assimilates the observations sequentially in time as they become available using the analysis step of the Kalman Filter, as illustrated in Fig. 1. The model trajectory is then discontinuous and, at a given analysis, the model is only informed by past and present observations. For the retrospective analysis of a time period in the past, *i.e.* a reanalysis, ensemble filters can easily be extended to smoothers to provide analyses that are informed by all past, present and future observations (Evensen and van Leeuwen , 2000; Li and Navon, 2001; Cosme et al., 2012; Nerger et al., 2014). Since the first version introduced by Evensen (1994) many variants have been developed mainly differing in the way the Kalman Filter analysis is rewritten and the analysed error covariance matrix resampled (e.g. Burgers et al., 1998; Houtekamer and Mitchell, 1998; Pham et al., 1998; Bishop et al., 2001; Nerger et al., 2012). A review of the most popular EnKFs using common notations can be found in Vetra-Carvalho et al. (2018). Efficient and parallel algorithms have been developed, and because they are independent of the forward model, several open-source toolboxes that implements various EnKFs are now available, e.g. OpenDA (https://www.openda.org ), PDAF (http://pdaf.awi.de).

As Monte-Carlo methods, EnKFs suffer from under sampling issues as often the size of the ensemble is much smaller than the size of the system to estimate. Localisation and inflation are popular methods to counteract these issues and to increase the stability of the filtering. Because they are based on the original Kalman Filter equations, EnKFs are optimal only for Gaussian distributions and linear models. However, the many applications in geoscience with large and non-linear models have shown that the method remains robust in general and EnKFs are used in several operational centres with atmosphere, ocean and hydrology models (e.g. Sakov et al., 2012; Houtekamer et al., 2009; Hendricks Franssen et al., 2011). While firstly developed for numerical weather and ocean prediction where the forecasts are very sensitive to the model initial state, the method is also widely used, *e.g.* in hydrology, for join state and parameters estimations (Sun et al., 2014).

In the context of ice-sheet modelling, encouraging results have been obtained by Bonan et al. (2014) for the estimation of the state and basal conditions of an ice-sheet model using the Ensemble Transform Kalman Filter (ETKF, Bishop et al., 2001; Hunt et al., 2007). They study the performance of the method using idealised twin experiments where perturbed observations generated from a model run are used in the DA framework to retrieve the true model states and parameters. Using a flowline shallow ice model, they show that both the basal topography and basal friction can be retrieved with good accuracy from surface observations with realistic noise levels, even for relatively small ensembles. The method has been further developed to assimilate the margin position in a shallow ice model that explicitly tracks the boundaries with a moving mesh method (Bonan et al., 2017).

The purpose of this paper is to explore the performance of ensemble Kalman filtering for the initialisation of a marine ice sheet model that includes GL migration. In particular, we want to address *(i)* the quality of the analysis for the simultaneous estimation of the basal topography and friction in the context of a marine ice sheet that is undergoing an unstable GL retreat, and *(ii)* the effects of the remaining uncertainties for the predictability of GL retreat. The ice flow model and the EnKF used in this study are described in Section 2. To test the DA framework, we define a twin experiment in Section 3. Section 4 presents the results for both the transient assimilation and the forecasts. Finally, perspectives and challenges for real applications are discussed in Section 5, before concluding remarks.

## 2 Methods

### 2.1 Ice flow model

The gravity-driven free surface flow of ice is solved using the finite-element ice flow model Elmer/Ice (Gagliardini et al., 2013).

For the force balance, we solve the shelfy-stream approximation (SSA) equation (MacAyeal, 1989) in one horizontal dimension. This is a vertically integrated model that derives from the Stokes equations for small aspect ratio and basal friction. In 1D, this leads to the following non-linear partial differential equation for the horizontal velocity field $u$

$$\frac{\partial}{\partial x}\left(4\bar{\eta}H\frac{\partial u}{\partial x}\right) - \tau_b = \rho_i g H \frac{\partial z_s}{\partial x} \tag{1}$$

whith $\rho_i$ the ice density, $g$ the gravity norm, $H = z_s - z_b$ the ice thickness with $z_s$ and $z_b$ the top and bottom surface elevations, respectively. Using the Glen's constitutive flow law, the vertically averaged effective viscosity $\bar{\eta}$ is given by

$$\bar{\eta} = \frac{1}{H}\int\limits_{z_b}^{z_s} \frac{1}{2}A^{-1/n}D_e^{(1-n)/n}\mathrm{d}z \tag{2}$$

where $D_e$ is the second invariant of the strain-rate tensor, equal here to $D_e^2 = (\partial u/\partial x)^2$, $A$ is the rate factor and $n$ is the creep exponent, taken equal to the usual value $n = 3$ in the following. The basal friction $\tau_b$ is null under floating ice and is represented with the non-linear Weertman friction law for grounded ice

$$\tau_b = Cu^m \tag{3}$$

with $C$ and $m$ the friction coefficient and exponent, respectively. In the following, we use the classical power law with $m = 1/n = 1/3$. When in contact with the ocean, the ice is assumed to be in hydrostatic equilibrium. The floating condition is evaluated directly at the integration points and $\tau_b$ in Eq. (1) is set to 0 wherever ice is floating (Seroussi et al., 2014).

The time dependency is introduced by the evolution of the top and bottom free surfaces. Because of the hydrostatic equilibrium, the ice sheet topography is fully defined by the bed elevation $b$ and only one prognostic variable. Equation (1) is then coupled with the vertically integrated mass conservation equation for the evolution of the ice thickness $H$

$$\frac{\partial H}{\partial t} + \frac{\partial (uH)}{\partial x} = a_s - a_b \tag{4}$$

with $a_s$ the surface accumulation rate and $a_b$ the basal melt rate. The free surfaces $z_s$ and $z_b$ are obtained from the floating condition which, for $z_s$, using a constant sea level $z_{sl} = 0$, gives

$$\begin{cases} z_s = b + H & \text{for } H \geq -b\dfrac{\rho_w}{\rho_i} \\ z_s = H\left(1 - \dfrac{\rho_i}{\rho_w}\right) & \text{otherwise} \end{cases} \tag{5}$$

with $\rho_w$ the sea water density.

## 2.2 Data Assimilation

### 2.2.1 Filter Algorithm

For the assimilation, we use the Error Subspace Ensemble Transform Kalman Filter (ESTKF, Nerger et al., 2012). Originally derived from the singular evolutive interpolated Kalman filter (SEIK, Pham et al., 1998), ESTKF leads to the same ensemble transformations as the ETKF but at a slightly lower computational cost. In practice we use the local version of the filter implemented in PDAF (http://pdaf.awi.de Nerger et al., 2005b) and coupled to Elmer/Ice in an offline mode. This section outlines the ESTKF algorithm.

As an EnKF, ESTKF approximates the state $\boldsymbol{x}^k$ and the error covariance matrix $\mathbf{P}_k$ of a system at time $t_k$ using an ensemble of $N_e$ realisations $\boldsymbol{x}_i^k$, $i = 1, ..., N_e$. The state vector, of size $N_x$, contains the prognostic variables and model parameters to be estimated and is approximated by the ensemble mean

$$\bar{\boldsymbol{x}}^k = \frac{1}{N_e} \sum_{i=1}^{N_e} \boldsymbol{x}_i^k \tag{6}$$

while the error covariance matrix is approximated by

$$\mathbf{P}_k = \frac{1}{N_e - 1} \mathbf{X}'_k \mathbf{X}'^T_k \tag{7}$$

where $\mathbf{X}'_k = (\boldsymbol{x}_1^k - \bar{\boldsymbol{x}}^k, ..., \boldsymbol{x}_{N_e}^k - \bar{\boldsymbol{x}}^k) \in \mathbb{R}^{N_x \times N_e}$ is the ensemble perturbation matrix.

The algorithm can be decomposed in two steps, the *forecast* and the *analysis*. Superscripts $^f$ (resp. $^a$) denote quantities related to each step respectively. The *forecast* propagates the state and the error covariance matrix of the system forward in

time, from a previous analysis at $t = t_{k-1}$ to the next observation time $t = t_k$. For this, the numerical model $\mathcal{M}_k$, assumed perfect in the sequel, is used to propagate each ensemble member individually during $n_{dt}$ model time steps

$$\boldsymbol{x}_i^{f,k} = \mathcal{M}_k(\boldsymbol{x}_i^{a,k-1}) \tag{8}$$

At $t = t_k$, a vector of observations $\boldsymbol{y}_o^k$ of size $N_y$ (with usually $N_y << N_x$) is available. $\boldsymbol{y}_o^k$ is related to the true system state $\boldsymbol{x}^f$ by $\boldsymbol{y}_o^k = \mathcal{H}(\boldsymbol{x}^f) + \boldsymbol{\epsilon}^k$ where the observation error $\boldsymbol{\epsilon}^k$ is assumed to be a white Gaussian distributed process with known covariance matrix $\mathbf{R}^k$, and $\mathcal{H}$ is the observation operator that relates the state variables to the observations. When $\boldsymbol{y}_o^k$ is the observed surface velocities, the relation between the observations and the system state, *i.e.*, the ice-sheet geometry, and ,parameters, *i.e.* the boundary conditions, is given by the force balance equation (1), thus $\mathcal{H}$ is a non-linear elliptic partial differential equation.

The *analysis* provides a new estimation of the system state by combining the information from the forecast and the observations. In the following we will omit the time index $^k$ in the notations as all the analysis is performed at $t = t_k$. As others EnKFs, ESTKF uses the Kalman Filter update equations to compute the analysed system state $\bar{\boldsymbol{x}}^a$ and covariance matrix $\mathbf{P}^a$ from the forecast, the observations and their uncertainties:

$$\begin{cases} \bar{\boldsymbol{x}}^a = \bar{\boldsymbol{x}}^f + \mathbf{K}\boldsymbol{d} \\ \mathbf{P}^a = (\mathbf{I} - \mathbf{K}\mathbf{H})\mathbf{P}^f \end{cases} \tag{9}$$

where $\boldsymbol{d} = \boldsymbol{y}_o - \mathcal{H}(\bar{\boldsymbol{x}}^f)$ is the *innovation* and $\mathbf{K}$ is the *Kalman gain* given by

$$\mathbf{K} = \mathbf{P}^f \mathbf{H}^T (\mathbf{H}\mathbf{P}^f\mathbf{H}^T + \mathbf{R})^{-1} \tag{10}$$

Here, $\mathbf{H}$ is the linearised observation operator at the forecast mean. However, in practice $\mathbf{H}$ does not need do be computed as it always acts as an operator to project the ensemble members in the observation space. Defining the forecast ensemble projected in the observation space by $\boldsymbol{y}_i^f = \mathcal{H}(\boldsymbol{x}_1^f)$, $i = 1, ..., N_e$ with $\bar{\boldsymbol{y}}^f$ the ensemble mean, we make the linear approximation

$$\mathbf{Y}^f = \mathbf{H}\mathbf{X}^f \tag{11}$$

with $\mathbf{X}^f = (\boldsymbol{x}_1^f, ..., \boldsymbol{x}_{N_e}^f) \in \mathbb{R}^{N_x \times N_e}$ the forecast ensemble matrix and $\mathbf{Y}^f = (\boldsymbol{y}_1^f, ..., \boldsymbol{y}_{N_e}^f) \in \mathbb{R}^{N_y \times N_e}$ its equivalent in the observation space.

In practice, with large models ($N_x >> 1$), the covariance matrices $\mathbf{P}^f$ and $\mathbf{P}^a$ of size $N_x \times N_x$ can not be formed, so that, to be implemented, the analysis (Eq. 9) needs to be reformulated. Moreover, the sample covariance matrix approximated with an ensemble of size $N_e$ (Eq. 7) is only a low-rank approximation of the true covariance matrix and its rank is at most $N_e - 1$. ESTKF uses this property to write the analysis in a $(Ne - 1)$-dimensional subspace spanned by the ensemble and referred to as the error subspace (Nerger et al., 2005a). The forecast covariance matrix $\mathbf{P}^f$ is then rewritten as

$$\mathbf{P}^f = \frac{1}{N_e - 1} \mathbf{L}\mathbf{L}^T \tag{12}$$

where $\mathbf{L} \in \mathbb{R}^{N_x \times N_e - 1}$ is given by

$$\mathbf{L} = \mathbf{X}^f \mathbf{\Omega} \tag{13}$$

The matrix $\mathbf{\Omega} \in \mathbb{R}^{N_e \times N_e - 1}$ defined as

$$\Omega_{ij} = \begin{cases} 1 - \dfrac{1}{N_e} \dfrac{1}{\frac{1}{\sqrt{N_e}} + 1} & \text{for } i = j,\ i < N_e \\[3mm] -\dfrac{1}{N_e} \dfrac{1}{\frac{1}{\sqrt{N_e}} + 1} & \text{for } i \neq j,\ i < N_e \\[3mm] -\dfrac{1}{\sqrt{N_e}} & \text{for } i = N_e \end{cases} \tag{14}$$

projects the ensemble matrix $\mathbf{X}^f$ onto the error subspace. The multiplication with $\mathbf{X}^f$ subtracts the ensemble mean and a fraction of the last column of the ensemble perturbation matrix $\mathbf{X}'^f$ from all other columns.

    After some algebra using Eq. (12) and Eq. (9), $\mathbf{P}^a$ can be written as a transformation of $\mathbf{L}$

$$\mathbf{P}^a = \mathbf{L} \mathbf{A} \mathbf{L}^T \tag{15}$$

with the transform matrix $\mathbf{A} \in \mathbb{R}^{N_e - 1 \times N_e - 1}$ given by

$$\mathbf{A}^{-1} = \rho(N_e - 1)\mathbf{I} + (\mathbf{Y}^f \mathbf{\Omega})^T \mathbf{R}^{-1} \mathbf{Y} \mathbf{\Omega} \tag{16}$$

where $\rho \in [0, 1]$ is the *forgetting factor* discussed in section 2.2.2.

    Finally, the update step is obtained as a single equation for the transformation of the forecast ensemble $\mathbf{X}^f$ to the analysed ensemble $\mathbf{X}^a$ as

$$\mathbf{X}^a = \bar{\mathbf{X}}^f + \mathbf{X}^f \mathbf{\Omega} (\bar{\mathbf{W}} + \mathbf{W}) \tag{17}$$

where $\bar{\mathbf{X}}^f$ is the matrix where the columns are given by the forecast ensemble mean, $\bar{\mathbf{W}}$ is a matrix where the columns are given by the vector

$$\bar{w} = \mathbf{A}(\mathbf{Y}\mathbf{\Omega})^T \mathbf{R}^{-1}(\boldsymbol{y}_o - \bar{\boldsymbol{y}}^f) \tag{18}$$

and $\mathbf{W}$ is given by

$$\mathbf{W} = \sqrt{N_e - 1}\, \mathbf{C} \mathbf{\Omega}^T \tag{19}$$

where $\mathbf{C}$ is the symmetric square root of $\mathbf{A}$ obtained by singular value decomposition.

    Finally, the analysed ensemble $\mathbf{X}^a$ is used as the initial ensemble for the next forecast, and so on up to the end of the data assimilation window.

    We draw attention to several remarks on the algorithm:

- To compute the innovation $\boldsymbol{d}$, we have made the same linear approximation $\mathcal{H}(\bar{\boldsymbol{x}}^f) = \bar{\boldsymbol{y}}^f$ as Hunt et al. (2007). This choice is consistent with the computation of the covariance matrices $\mathbf{P}^f\mathbf{H}^T$ and $\mathbf{H}\mathbf{P}^f\mathbf{H}^T$ in Eq. (10) using the linear approximation Eq. (11) (Houtekamer and Mitchell, 2001).

- Several ensembles can have the same mean and covariance matrix, this is why several EnKFs exactly satisfy Eq. (9) but lead to different ensemble transformations and thus different analysed ensembles (Vetra-Carvalho et al., 2018). With the same arguments several variants of ESTKF can be introduced, *e.g.* by replacing $\mathbf{\Omega}$ in Eq. (19) by a random matrix with the same properties or using a Cholesky decomposition to compute $\mathbf{C}$.

- As written here, the ESTKF leads to the same ensemble transformation as the ETKF. However, as the computations are not performed in the same sub-space tiny differences due to the finite precision of the computations may grow leading to slight differences at the end of the assimilation window (Nerger et al., 2012).

- The leading computational cost of the ensemble transformation in ESTKF is $\mathcal{O}\left(N_y(N_e-1)^2 + N_e(N_e-1)^2 + N_xN_e(N_e-1)\right)$, so it scales linearly with $N_x$ and $N_y$ (Nerger et al., 2012). Naturally, increasing $N_e$ also requires to increase the number of model runs and, in general, the objective is to get the ensemble size as small as possible. The performance of the algorithm also depends on the evaluation of the product of $\mathbf{R}^{-1}$ with some vectors, which can become more expensive when the observation errors are spatially correlated.

### 2.2.2 Filter stabilisation: inflation and localisation

In practice for large scale problems, EnKFs as Monte-Carlo methods suffer from under-sampling issues. First, because of the rank deficiency of the covariance matrix $\mathbf{P}^f$, the analysis adjusts the model state only in the error subspace, ignoring error directions not accounted for by the ensemble (Hunt et al., 2007). This can result in an analysis that is overconfident and underestimates the true variances. On the long run, the ensemble spread will become too small and the analysis will give to much weight on the forecast finally disregarding the observations and diverging from the true trajectory. A common simple *ad-hoc* remedy is to inflate the forecast covariance matrix with a multiplicative factor (Pham et al., 1998; Anderson and Anderson, 199). Here, inflation has been introduced in Eq. (16) using the forgetting factor $\rho \in [0,1]$ with $\rho = 1$ corresponding to no inflation (Pham et al., 1998). It is the inverse of the inflation factor used by Bonan et al. (2014).

Second, the rank deficiency of $\mathbf{P}^f$ leads to the appearance of spurious correlations between parts of the system that are far away. As these correlations are usually small, a common remedy is to damp these correlations with a procedure called localisation. In covariance localisation, localisation is applied by using an ensemble covariance matrix that results from the Schur product of $\mathbf{P}^f$ with an *ad-hoc* correlation matrix that drops long range correlations (Hamill et al., 2001; Houtekamer and Mitchell, 2001). However, this localisation technique is not practical for square-root filters where $\mathbf{P}^f$ is never explicitly computed. Here, as in Bonan et al. (2014), we use a localisation algorithm based on domain localisation and observation localisation (Ott et al., 2004; Hunt et al., 2007). Both methods are illustrated in Sakov and Bertino (2011) who conclude that they yield to similar results. Domain localisation assumes that observations far from a given location have negligible influence. In practice, the state vector in each single mesh node is updated independently during a loop through the nodes that can easily

be parallelized for numerical efficiency. For each local analysis, only the observations within a given radius $r$ from the current node are used. In addition to avoid an abrupt cut-off, the observation error covariance matrix $\mathbf{R}$ is modified so that the inverse observation variance decreases to zero with the distance from the node using a fifth-order polynomial function which mimics a Gaussian function but has compact support (Gaspari and Cohn, 1999). Because it drops spurious long-range correlations and allows the local analyses to choose different linear combinations of the ensemble members in different regions, localisation implicitly increases the rank of the covariance matrix, leading to a larger dimension of the error subspace, implicitly increasing the effective ensemble size and the filter stability (Nerger et al., 2006; Hunt et al., 2007). However, it has been reported that localisation could produce imbalanced solutions (Mitchell et al., 2002). Here, because the force balance are non-inertial and the SSA assumes that the ice-shelves are in hydrostatic equilibrium, this should not be an issue. Another disadvantage is that, when long-range correlations truly exist, the analysis will ignore useful information that could have been used from distant observations.

Here, the forgetting factor $\rho$ and the localisation radius $r$ will be used as tuning parameters of the filter. Improving the theoretical understanding of these *ad hoc* procedures and developing adaptive scheme is an active research area and interested readers can refer to review articles (e.g. Bannister, 2017; Carrassi et al., 2018; Vetra-Carvalho et al., 2018).

## 3 Experimental design

To evaluate the performance of the DA framework we perform a twin experiment. In this section we first describe the synthetic reference simulation that will be used to assess the performance of the DA framework. From this reference, we generate a set of synthetic noisy observations that will be used by the assimilation scheme. Finally, we describe the initial ensemble constructed using *a priori* or background information.

### 3.1 Reference simulation

We start by building an initial steady marine ice-sheet. The domain extends from $x = 0\,\mathrm{km}$ where we apply a symmetry condition, $u = 0$ in Eq. (1), to $x = 800\,\mathrm{km}$ where we have a fixed calving front. We use 1D linear elements with a uniform mesh resolution of $200\,\mathrm{m}$, leading to 4001 mesh nodes.

Following Durand et al. (2011), we generate a synthetic bed geometry that reproduces a typical large-scale overdeepening with some small scale roughness. The bed $b = b_{trend} + b_r$ is the sum of a general trend $b_{trend}$ defined as

$$b_{trend} = \begin{cases} -1100 + x & \text{for } x \leq 450\,\mathrm{km} \\ -650 - 5(x - 450) & \text{for } x > 450\,\mathrm{km} \end{cases} \tag{20}$$

and a roughness signal $b_r$ that is computed at $200\,\mathrm{m}$ resolution using a random midpoint displacement method (Fournier et al., 1982). This is a classical algorithm for artificial landscape generation. In 1D, the algorithm recursively subdivide a segment and a random value drawn from a normal distribution $\mathcal{N}(0, \sigma^2)$ is added to the elevation of the midpoint. The standard deviation

$\sigma$ is decreased by a factor $2^h$ between two recursions. Here we have used 12 recursions using an initial standard deviation $\sigma = 500\,\mathrm{m}$ and a roughness $h = 0.7$. The resulting bed is shown in Fig. 2.

For the basal friction, we use a synthetic sinusoidal function with two wavelengths for $C$ ($\mathrm{MPa\,m}^{-\frac{1}{3}}\,\mathrm{a}^{-\frac{1}{3}}$)

$$C = 0.020 + 0.015\sin(5\frac{2\pi x}{L})\sin(100\frac{2\pi x}{L}) \tag{21}$$

5  with $L = 800\,\mathrm{km}$ (Fig. 3).

While not tuned to match any specific glacier, this synthetic design compares relatively well to the conditions found in Thwaites Glacier (Antarctica). Thwaites has been the focus of many recent studies as it is undergoing rapid ice loss and, connected to deep marine-based basins, its retreat could trigger a large scale collapse of the West Antarctic Ice Sheet over the next centuries (Scambos et al., 2017). In Fig. 4, $C$ and $b$ are compared with model results from Brondex et al. (2018) along three streamlines. In Brondex et al. (2018) , $C$ has been inferred from the observed surface velocities using a time-independent control inverse method and a SSA model. We can see that our synthetic design is realistic both in terms of amplitude and spatial variations. As the other characteristics (geometry, small flow divergence/convergence) are also similar, the model velocities have the good order of magnitude.

Using an uniform ice rigidity $B = (2A)^{-1/n} = 0.4\,\mathrm{MPa\,a}^{\frac{1}{3}}$, we grow an ice sheet to steady state using a uniform surface accumulation $a_s = 0.5\,\mathrm{m\,a}^{-1}$ and no basal melting $a_b = 0$. The steady state GL is located at $x = 440\,\mathrm{km}$, just downstream of the region of overdeepening (Fig. 2).

In Jenkins et al. (2018), observed ice-flow accelerations in the Amundsen sea sector have been attributed to the decadal oceanic variability, where warm phases associated with increased basal melt induce a thinning of the ice shelves reducing their buttressing effect initiating short lived periods of unstable retreat of the most vulnerable GLs. In a flow line experiment the ice shelf do not exert any buttressing effect. Using a suite of melting and calving perturbation experiments for Pine Island Glacier, Favier et al. (2014) have shown that, when initiated, the dynamics of the unstable retreat is fairly independent of the type and magnitude of the perturbation. Here, to trigger the initial acceleration, we instantaneously decrease the ice rigidity to $B = 0.3\,\mathrm{MPa\,a}^{\frac{1}{3}}$ at $t = 0$, keeping all the other parameters constant.

This initial perturbation induce an acceleration, a thinning and a retreat of the GL. The model is then run for 200 years with a time step $dt = 5\,10^{-3}\,\mathrm{a}^{-1}$. After a short stabilisation at $x = 437.2\,\mathrm{km}$ between $t = 13\,\mathrm{a}$ and $t = 32\,\mathrm{a}$, the GL retreats at a rate of approximately $1\,\mathrm{km\,a}^{-1}$ during the following 100 years, then the rate decreases as the GL enters an area of down-slopping bed (Fig. 2). The retreat rate shows small variations associated with spatial variations of the topography and basal friction.

## 3.2  Synthetic Observations

From the reference run, we generate synthetic noisy observations that are typical of the resolution and performance of actual observing systems.

For the bed, we mimic an airborne radar survey conducted perpendicular to the ice flow with an along flow resolution of approximately $15\,\mathrm{km}$. For this, we randomly select 54 locations between $x = 0$ and $x = 800\,\mathrm{km}$, and then linearly interpolate the true bed and add a random uncorrelated Gaussian noise with a standard deviation $\sigma_b^{obs} = 20\,\mathrm{m}$ (Fig. 3).

We assume that the surface elevation and velocities are observed at an annual resolution at each mesh node. We then add an uncorrelated Gaussian noise with a standard deviation $\sigma_{z_s^{obs}} = 10\,\text{m}$ for the surface elevation and $\sigma_u^{obs} = 20\,\text{m}\,\text{a}^{-1}$ for the velocity. The most recent velocity products are now posted with a monthly to annual resolution (Mouginot et al., 2017; Joughin et al., 2018). The reported uncertainty for individual velocity estimates using the 6- and 12-day image pairs from the Sentinel 1A/B satellites is $6.2$ and $17.5\,\text{m}\,\text{a}^{-1}$ for the two horizontal velocity components in stable conditions; however this could be underestimated in the coastal areas. For the surface elevation, the spatial and temporal resolution as well as the coverage and uncertainty will depend on the sensors. The ArcticDEM (http://arcticdem.org) is a collection of openly available digital surface models derived from satellite imagery and posted at 2-m spatial resolution. After co-registration, a standard deviation ranging from 2 to $4\,\text{m}$ has been reported for the uncertainty of elevation difference between two individual models of static surfaces (Dai and Howat, 2017). Using the same satellites, Greenland digital elevation models are now posted with a 3-month temporal resolution (https://nsidc.org/data/nsidc-0715).

### 3.3 Assimilation setup

We recall that our aim is to initialise the model using the DA framework to estimate the state together with the basal conditions. As a simplification to realistic experiments, we assume in the following that the ice rheological properties (represented by the Glen flow law and its parameters) and the forcing (represented by the surface and basal mass balances in Eq. (4)) are perfectly known. In addition, we assume that the form of the basal friction follows Eq. (3) with $m = 1/3$, so that only the spatially-varying friction coefficient $C$ is uncertain.

In our model, as the force balance equation (1) contains no time derivative, the velocity is a diagnostic variable. Because of the flotation condition, the topography can be represented by only one prognostic variable. The state vector $\boldsymbol{x}$ is then given by the free surface elevation $z_s$ at every mesh node, and we use the floatation Eq. (5) for the mapping between the ice thickness $H$ and $z_s$. The state vector is augmented by the two parameters to be estimated, the bedrock topography $b$ and the basal friction coefficient $C$. For the parameters we assume a persistence model, *i.e.* no time evolution, during the forecast step (Eq. 8). Because the velocities are insensitive to the basal conditions where ice is floating, these two parameters are included in the state vector only for the nodes where at least one member is grounded. In addition, to insure that $C$ remains positive, we use the following change of variable for the assimilation $C = \alpha^2$. Although it does not insure uniqueness of the estimation as $\alpha$ and $-\alpha$ would lead to the same $C$, this change of variable is classical (MacAyeal, 1993) and was chosen as the reference friction coefficient spans only one order of magnitude. Similar performances where found using the other classical change of variable $C = 10^\alpha$ as in Gillet-Chaulet et al. (2012).

Because both $z_s$ and $b$ are included in the state vector, the analysis does not conserve the ice sheet volume, neither for the ensemble mean and the individual members. However, as illustrated in Fig. 1, the estimation of $z_s$ and $b$, and thus of the ice thickness, should be improved at each analysis as more data are assimilated, and the final state is the best estimation provided by the filter knowing the model, all the observations during the assimilation window and their uncertainties. As mentioned in the introduction, if the main interest is an analysis of past volume changes, a smoother or a variational method might be more appropriate. The smoother extension of the ESTKF can be found in Nerger et al. (2014). Note however that, interestingly, if

we expect that the filter will improve the estimation of the ice thickness there is no guaranty in general that it will provide a better estimate of the total volume as an a priori with a totally different thickness distribution could lead, by compensation of the errors, to a perfect estimate of the true volume.

Kalman-based filters are based on the hypothesis of the independence between the background, *i.e. the initial ensemble*, and the observations that are used during the assimilation. As the synthetic bed observations will be used to construct the initial ensemble (cf next section), we assimilate only the surface elevation and velocity observations, every year from $t = 1\,\mathrm{a}$ up to $t = 35\,\mathrm{a}$. The observation operator $\mathcal{H}$ is a simple mapping for the surface elevation, and is given by the non-linear SSA equation (Eq. 1) for the surface velocities.

Finally, to illustrate the effect of the transient assimilation on model projections on time scales relevant for sea level projections, the analysed states at $t = 20\,\mathrm{a}$ and $t = 35\,\mathrm{a}$ are used to run deterministic and ensemble forecasts up to $t = 200\,\mathrm{a}$. The deterministic forecast uses the ensemble mean produced by the analysis while the ensemble forecast propagates the full ensemble.

## 3.4 Initial ensemble

For atmosphere and ocean models, the initial state is usually sampled from a climatology, either observed or from a model run. This method can not be used for the parameters and the initial ensemble must reflect the background and the estimation of its uncertainty, available *a priori* before the assimilation. Following previous studies (Gudmundsson and Raymond, 2008; Pralong and Gudmundsson, 2011; Bonan et al., 2014; Brinkerhoff et al., 2016), we assume that the initial distributions for $b$ and $C$ are Gaussian with a given mean and a prescribed covariance model. Furthermore we assume no cross-correlation between the initial $b$, $C$ and $zs$ and we draw the initial ensembles independently.

For $b$ and $C$, the initial samples are drawn using the R package *gstat* (Pebesma and Wesseling, 1998). As classical in geostatistics, the covariance model is prescribed using a variogram $\gamma(d)$ that is half the variance of the difference between field values as a function of their separation $d$. It is usually defined by two parameters, the sill $s$ that defines the semi-variance at large distances and the range $r_a$ which, for asymptotic functions, is defined as the distance where the $\gamma(r_a) = 0.95s$. The package *gstat* allows directly to draw simulations, i.e. random realisations of the field, from the prescribed spatial moments (Pebesma and Wesseling, 1998).

For the bed we use an exponential function

$$\gamma(d) = s(1 - e^{-\frac{3d}{r}}) \tag{22}$$

with $r_a = 50\,\mathrm{km}$ and $s = 4000\,\mathrm{m}^2$. We also add a nugget model defined by

$$\gamma(d) = \begin{cases} 0 & d = 0 \\ nug & d > 0 \end{cases} \tag{23}$$

with $nug = 200\,\mathrm{m}^2$. This model is meant to represent the bed measurement error. To draw the initial ensemble, the simulations are conditioned with the bed observations.This procedure gives an initial ensemble that is drawn from the posterior probability

distribution that would be obtained using ordinary Kriging with the same observations and variograms. The ensemble mean and spread for a 50-members ensemble are shown in Fig. 3 and the first three members are shown in Fig. 5. As expected, the ensemble spread increases with the distance from the observations. At the observation locations, the spread is controlled by the nugget. For the individual members, the nugget controls the small scale variability, resulting in a roughness larger than the reference. When averaged this roughness disappears, and the ensemble mean has a much smoother topography.

For the friction coefficient, we assume that we know the mean value $C_{mean} = 0.020 \, \mathrm{MPa} \, \mathrm{m}^{-\frac{1}{3}} \, \mathrm{a}^{-\frac{1}{3}}$ and draw unconditional simulations. For the spatial dependence, we use a Gaussian function $\gamma(d) = s(1 - e^{-3(\frac{d}{r})^2})$ for the variogram using a range $r_a = 2.5 \, \mathrm{km}$ and a sill $s = 8.10^{-5} \, \mathrm{MPa}^2 \, \mathrm{m}^{-\frac{2}{3}} \, \mathrm{a}^{-\frac{2}{3}}$. This results in initial ensemble members that have approximately the same maximal amplitude as the reference, as shown in Fig. 5.

For the free surface, we initialise all the members using the observed (noisy) free surface at $t = 0$. Doing so, we implicitly assume that the spread of the ensemble induced by the uncertain initial conditions at the first analysis is small compared to the spread induced by the uncertain parameters. This is motivated by the fact that divergence anomalies induced by uncertainties in model parameters can typically reach tens to hundreds of meters per years in fast flowing areas (Seroussi et al., 2011).

## 4 Results

### 4.1 Assimilation

To assess the performance of the DA in retrieving the basal conditions we compute the root-mean-square error (RMSE) between the analysed ensemble mean and the reference for both the bed and the friction coefficient, $\mathrm{RMSE}_b$ and $\mathrm{RMSE}_C$ respectively. After each analysis, the RMSE is computed using all the nodes where the basal conditions have been updated by the assimilation, i.e. at least one member is grounded, and where $x \geq 300 \, \mathrm{km}$. The later value is close to the position reached by the grounding line after 200 years in the reference simulation, moreover, during the assimilation window, the reference velocity at this location is close to $80 \, \mathrm{m} \, \mathrm{a}^{-1}$ (Fig. 6), so that the relative noise is $\sim 25\%$ and we don't expect too much improvement from the DA upstream as the velocity tends to 0.

Here the size of the state vector $\boldsymbol{x}$, $N_x$, is approximately 8400, i.e. $z_s$ at every node and the basal conditions, $b$ and $C$, in the grounded part. To test the performances of DA in conditions that would be numerically affordable for real applications, we run the assimilation with relatively small ensemble sizes $N_e = 30$, $N_e = 50$ and $N_e = 100$. In this case, inflation and localisation are required to counteract the effects of undersampling and we test a range of forgetting factors $\rho$ and localisation radius $r$. The errors obtained at $t = 20 \, \mathrm{a}$ relative to the errors from the initial ensemble mean are shown in Fig. 7. The performances of the assimilation for $N_e = 50$ and $N_e = 100$ are very similar. The filter diverges and produces errors larger than the initial errors for a localisation radius $r \leq 4 \, \mathrm{km}$. However, for larger localisation radii, the assimilation is relatively robust for a wide range of $r$ and $\rho$, with errors reduced by $\sim 30\%$ for $b$ and $\sim 40\%$ for $C$. Decreasing the ensemble sizes reduces the filter performance but there is still a reduction of the errors by $\sim 20\%$ and $\sim 30\%$, respectively, with $N_e = 30$. For the two smallest ensembles, there is an optimal value for $r$ and increasing $r$ above this value decreases the filter performance. In general, this optimal value for $r$

increases as $\rho$ decreases, because the ensemble spread reduction induced by assimilating more observations is counterbalanced by the inflation.

In the sequel we discuss the results obtained with an ensemble size $N_e = 50$. As a compromise between the performances in retrieving $b$ and $C$, we choose a forgetting factor $\rho = 0.92$ and a localisation radius $r = 8\,\mathrm{km}$. The evolution of the RMSEs as a function of assimilation time together with the initial and final ensembles are shown in Fig. 3. $\mathrm{RMSE}_b$ decreases steadily from $\sim 25\,\mathrm{m}$ for the initial ensemble at $t = 0$ to $\sim 12\,\mathrm{m}$ at $t = 35\,\mathrm{a}$. For the basal friction, $\mathrm{RMSE}_C$ is decreased by a factor 1.75 during the first ten years, then there is still a slight but much smaller improvement as new observations are assimilated.

At the end of the assimilation, for both fields, the spatial variations are well reproduced by the ensemble mean and, compared to the initial ensemble, the difference from the reference is decreased everywhere except between $300$ and $325\,\mathrm{km}$ for $C$. The reduction in the error is also accompanied by a diminution of the ensemble spread, represented by the minimum and maximum values in Fig. 3. This reduction is the most important just upstream of the grounding line where the relative noise for the velocity is the smallest. For the first $100\,\mathrm{km}$ upstream of the grounding line, the ensemble standard deviation increases by a factor 4, from approximately $4$ to $17\,\mathrm{m}$ for $b$ and from $1.10^{-3}$ to $4.10^{-3}\,\mathrm{MPa}\,\mathrm{m}^{-\frac{1}{3}}\,\mathrm{a}^{-\frac{1}{3}}$ for $C$. Downstream of the GL where all members are floating, the model is insensitive to the basal conditions and the initial ensemble is unchanged.

We expect that uncertainties in the ice-sheet interior should not affect short-term forecast of the coastal regions (Durand et al., 2011), however for completeness we also show the results for the first $300\,\mathrm{km}$ in Fig. 8. For the bed there is only a small improvement of the ensemble mean with an RMSE decreasing from $50\,\mathrm{m}$ to $45\,\mathrm{m}$ after 35 years. Because the relative observation error on the velocity is very high in the first kilometres, the reduction of the ensemble spread due to the assimilation of new observations is very small and eventually outperformed by the inflation leading to an ensemble spread that becomes larger than before the assimilation. The model seems more sensitive to the basal friction and this effect is less pronounced for $C$ with a continuous decrease of the RMSE and a small reduction of the ensemble spread everywhere.

Figure 2 shows that some members undergo a fast GL retreat of few kilometres before assimilation at the end of the first year. Interestingly, as the assimilation updates both the thickness and the bed, it also corrects the GL position which never departs by more than few nodes from the reference for the rest of the assimilation period.

As in realistic simulations, the true bed and friction are not available to assess the performance of the DA, we also look at the variables assimilated by the model. Figure 9 shows the RMSEs between the ensemble mean and the reference for the velocity $u$ ($\mathrm{RMSE}_u$) and the free surface $zs$ ($\mathrm{RMSE}_{z_s}$), computed for the entire domain ($0 \leq x \leq 800\,\mathrm{km}$). We also report the evolution of the ensemble spread, computed as the square root of the averaged ensemble variance. The velocities before and after the analysis at $t = 1\,\mathrm{a}$ and $t = 35\,\mathrm{a}$ are shown in Fig. 6. The RMSEs are largely decreased during the first few years, especially for the velocity with an error of more than $300\,\mathrm{m}\,\mathrm{a}^{-1}$ before the first assimilation to approximately the noise level, $20\,\mathrm{m}\,\mathrm{a}^{-1}$, at $t = 20\,\mathrm{a}$. For $z_s$, $\mathrm{RMSE}_{z_s}$ is already below the noise level before the first analysis and decreases relatively steadily to reach $\sim 2\,\mathrm{m}$ after 35 years. $\mathrm{RMSE}_u$ increases at the end of the period when the reference GL leaves the stable region. As can be shown in Fig. 6, the error is dominated by the larger difference over the ice-shelf due to the few members that still have their GL at the stable location, largely affecting the ensemble mean.

In general, during the first and last years of the assimilation period, the error and the ensemble spread increase during the forecast step. The analysis step reduces both the error and the ensemble spread (Fig. 9). With the stabilisation of the grounding line, both the error and the spread remain relatively stable during the forecast, and as $RMSE_u$ and $RMSE_{z_s}$ have already reached levels comparable to the observation noise, there is no much improvement during the analysis. After few assimilation steps, as expected for a reliable ensemble, the error and the spread have similar values.

Similar conclusions are drawn if the assimilation is pursued up to $t = 50\,\mathrm{a}$. Because of the sensitivity of the ice-shelf velocities to the grounding line position, $RMSE_u$ show a higher variability, but expect for few exceptions, stay close to the noise level. $RMSE_b$ and $RMSE_c$ stagnate as continue to improve the reconstruction mostly in the first few tens of kilometres upstream of the GL where the relative noise on $u$ is the smallest.

To asses the influence of the observation uncertainties in the performance of the DA, we repeat the experiment with the same localisation and inflation but different levels for the uncertainties on the observed surface velocity ($\sigma_u^{obs}$) and surface elevation ($\sigma_{z_s}^{obs}$) (cf section 3.2). We recall that these uncertainties are not correlated spatially and temporally. As shown in Fig. 10, the performance of the DA to retrieve both $b$ and $C$ increases when the uncertainty on the velocity observation $\sigma_u^{obs}$ decreases. However, when looking at the model velocities and surface elevation, this improvement is not significant as the RMSEs were already below the noise level. As shown in Fig. 11, as expected, decreasing $\sigma_{z_s}^{obs}$ improves the analysis for the surface elevation. However, it does not necessarily reflect on the basal conditions and, on the contrary, reducing $\sigma_{z_s}^{obs}$ below $10\,\mathrm{m}$ leads to an increase of $RMSE_C$ from $0.004$ to $0.005\,\mathrm{Mpa\,m^{-1/3}\,a^{1/3}}$ . However, again, this effect does not reflect on the model velocities that are retrieved with the same accuracy.

## 4.2  Forecast simulations

We now discuss model projections from the initial state to $t = 200\,\mathrm{a}$.

Without assimilation, the deterministic forecast, i.e. using the ensemble mean basal conditions, rapidly leads to the fastest GL retreat and, after few years the GL position is no more included within the previsions from the ensemble (Fig.2B). This is due to the fact that the ensemble mean is smoother than the reference and any of the ensemble members. The reference GL position is included in the ensemble and at the end of the simulations most of the members are within $\pm 25\,\mathrm{km}$ from the reference. However, for few members the GL remains very stable near its initial position for tens to hundreds of years, eventually never switching to an unstable regime during the duration of the simulation. Retreat rates are relatively variable from one member to the other, depending on the basal conditions.

With assimilation, the ensemble mean is improved and the difference from the reference reduced. The deterministic forecast cannot be distinguished from the ensemble members any more (Fig.2C-D). Retreat rates are closer to the reference, with the previsions from all the ensemble members being more or less parallel to the reference. We note however that, when the forecast starts after an assimilation window of 20 years, i.e. during a period of stable GL position for the reference, the deterministic forecast leaves the stable position with a delay of approximately 25 years, and a few members remain stable for the entire simulation. On average between $t = 13$ to $t = 32\,\mathrm{a}$, the thinning rate at the GL in the reference simulation, is approximately 0.6 m/a, reducing to 0.25 m/a during the last two years. The total thinning between two analyses is then much lower than the noise

in the observed surface elevation and cannot be captured accurately by the DA. In addition, at the GL, the difference between the minimum and maximum bed elevation given by the ensemble is approximately 20 meters. This remaining uncertainty induces a difference of more than two meters for the floatation surface, and combined with the small thinning rates explains the delays in the initiation of the instability.

Extending the assimilation window up to $t = 35\,\mathrm{a}$ when the reference has switched in a fast retreat, allows to force all the members in the unstable retreat. There is a very good agreement between the reference and the deterministic forecast up to $t = 110\,\mathrm{a}$. This is also true for the ensemble and after that the spread is larger and the predicted GLs are less retreated than for the reference.

     These results can be summarized by looking at the distribution of the ensemble forecasts for the grounding line position
and volume above floatation (VAF) at $t = 100\,\mathrm{a}$ in Fig. 12 where the relative VAF change is computed as $(VAF_{t=100} - VAF_{t=0}^{ref})/VAF_{t=0}^{ref}$ with $VAF_{t=0}^{ref}$ the reference VAF at $t = 0$. As expected there is a clear correlation between grounding line retreat and mass loss, higher retreat leading to higher mass loss. The distributions are clearly non Gaussian, however, even without assimilation there is already a mode close to the reference. The mode is more pronounced, with more members close to the reference as observations are assimilated. As discussed before, with no assimilation or a short assimilation up to $t = 20\,\mathrm{a}$
before the unstable retreat, the deterministic forecast can be very different from the mode of the ensemble forecast. However they are very similar if the assimilation is pursued up to $t = 35\,\mathrm{a}$, within $1\%$ for the relative volume loss or 5km for the GL position.

## 5   Discussion

Here, we have tested an ensemble Kalman Filter to assimilate annually observed surface velocities and surface elevation in a
marine ice-sheet model. Similarly to previous studies, we have shown that, in fast flowing regions, it is possible to accurately separate and recover both the basal topography and basal friction from surface observations (Gudmundsson and Raymond, 2008; Goldberg and Heimbach, 2013; Bonan et al., 2014; Mosbeux et al., 2016). In view of our results, because the synthetic bed observations were already used once to generate the initial ensemble, it seems unnecessary to assimilate these same observations again during each analysis as in Bonan et al. (2014).

Using a scheme that assimilates time-dependent observations provides a model state consistent with transient changes and that can directly serve as an optimal initial condition to run forecast simulations without the need of an additional relaxation (Goldberg and Heimbach, 2013; Goldberg et al., 2015). Interestingly the position of the grounding line is also corrected during the analysis step, and the ensemble quickly converges within few grid nodes from the reference. In addition, the ensemble framework naturally allows to estimate and propagate the uncertainty of the estimated parameters. Each assimilation of new
data improves the reconstruction of the basal conditions (Fig. 3), and the first ten years are very efficient in reducing error and the spread of the model surface velocities and elevation (Fig. 9). Furthermore, we have shown that the remaining uncertainties in the basal conditions do not significantly affect GL retreat rates once the unstable retreat is engaged. However, they can lead to considerable delays in the initiation of the instability. If the assimilation is pursued up to the beginning of the instability (35

years in our experiment) all the members exhibit the instable retreat and centennial-scale model projections converge to the reference (Fig. 12).

Good results have been obtained with relatively small ensembles (50 to 100 members) for a state vector of size $N_x \approx 8400$ and $N_y = 4002$ observations. Similarly to Bonan et al. (2014), we still see an improvement with a 30-members ensemble but the performances to retrieve the basal conditions are not as good. Running 2D plane view simulations with such ensemble sizes is largely possible as demonstrated by Ritz et al. (2015) who, using an hybrid shallow ice-shallow shelf model, have run a 200 years ensemble forecast of the whole Antarctic Ice Sheet using 3000 members.

We have used inflation and localisation to stabilise the filter. The inflation giving the best results in Bonan et al. (2014) ($\rho = 0.87 - 1.02$) is similar to the values tested in this study. For the localisation radius $r$ we have used values between 4 and 16 km, while it ranges from 80 to 120 km in Bonan et al. (2014). While this seems counter-intuitive as the velocities depends only on the local conditions with the shallow ice approximation used by Bonan et al. (2014), in fact, because we use a different grid size ($dx = 0.2$km compared to $dx = 5$km in Bonan et al. (2014)), for each node we assimilate twice as much observations. Our results are in agreement with the adaptive localisation radius proposed by Kirchgessner et al. (2014). Using three different models, Kirchgessner et al. (2014) have shown that good performances are obtained when $r$ is such that the effective local observation dimension, defined as the sum of the weights attributed to each observation during the local assimilation, is equal to the ensemble size. Here the observation weights decrease with the distance to the local assimilation domain following a fifth-order polynomial function mimicking a Gaussian function (Gaspari and Cohn, 1999). The value $r = 8$km used for the 50 members-ensemble gives an effective observation dimension of 56. Future studies should investigate if this result can be transposed to realistic 2D simulations with unstructured meshes.

In the experiments presented above, we have used a depth integrated model for the force balance equations where GL migration is implemented through a hydrostatic floatation condition. This allows to fully describe the ice topography with only one prognostic variable. Adaptation of the framework to a full-Stokes model requires minimum adaptations, however these models do not rely on the floatation condition and solve a proper contact problem for the grounding line migration (Durand et al., 2009), this implies to incorporate the two prognostic free surfaces $z_b$ and $z_s$ in the state vector. These models might be more sensitive to unbalanced geometries that could result from the analyses, especially when localisation is used (Cohn et al., 1998; Houtekamer and Mitchell, 2001). However, the ESTKF, as the ETKF, induces a minimal transformation of the ensemble members and thus has better chances to preserve balance (Nerger et al., 2012).

Before generalizing such methods to real glacial systems, several points must be taken in consideration. They are independent of the DA method but they will eventually be treated differently in a variational or in an ensemble framework.

First, if the implementation is not an issue, the computational cost implied by running a full-Stokes model might remain a limiting factor. Compared to the Stokes solution, the SSA is known to overestimates the effects of bed topography perturbations on the surface profile for wavelengths less than few ice thicknesses (Gudmundsson, 2008). How this issue can affect the reconstruction of the basal properties has never been quantified, however snapshot basal friction inversions have shown that the solution is sensitive to the force balance approximation (Morlighem et al., 2010). In addition, the MISMIP experiments have shown that the GL position and its response to a perturbation depend on the force balance solved by the models (Pattyn et al.,

2012, 2013). In real applications, the performance of DA can be improved by explicitly taking into account the model error. Several strategies have been developed to account for this error, one approach with EnKFs being to use different versions of the model for different ensemble members (Houtekamer et al., 2009). Further studies could investigate the potential benefits of using ensembles that combine several force balance approximations.

5  Second, the quality of the analysis and the accuracy of the error estimates depends on the observation error covariance matrix $\mathbf{R}$. It is then important to provide meaningful error estimates. Recent velocity maps provide an error estimates reported as the $1\sigma$ value for each individual location (Mouginot et al., 2012; Joughin et al., 2018). In general, this value agrees well with independent estimates, however care must be taken when the maps results from a composite of different sensors or different periods and in general it might be difficult to properly estimate $\mathbf{R}$.

10  In a review paper Tandeo et al. (2018) illustrate the impacts of badly calibrated observation and model error covariance matrices in a sequential DA framework and discuss available methods and challenges for their joint estimation. For the question of the impact of systematic errors, *i.e.* bias, either in the model and in the observations, and their correction by augmenting the system state in variational and ensemble DA, interested readers are referred to Dee (2005).

  Third, the results depends on prior assumptions on the control variables and their variability, represented here by the initial
15 ensemble. For the basal topography, current reference maps provide local error estimates (Fretwell et al., 2013; Morlighem et al., 2017), however they do not provide information about spatial correlations so that generating initial ensembles with the correct statistics might be problematic. In addition, the gridding can result in a loss of information for some regions of dense measurements, or can lead to too smooth terrains in sparsely sampled areas. With the aim of generating terrains that have the correct high-resolution roughness, Graham et al. (2017) propose a synthetic 100-m resolution Antarctic bed elevation
20 that combines the reference topography bedmap2 (Fretwell et al., 2013) with an unconditional simulation where the spatial correlation is fitted from dense radar measurements. This method could be used to generate initial ensembles but requires to have access to the initial high resolution measurements. Generating initial ensembles for the basal friction might be more problematic as there is in general no independent *a-priori* information about the magnitude and spatial variability of the basal friction. If there is a correlation between the basal drag and the seismic observations of the bed conditions at large scale, a
25 proper physical theory is still missing to quantitatively incorporate such information in the models (Kyrke-Smith et al., 2017). It could be interesting to investigate how the existing multi-model basal friction reconstructions, based on snapshot inversions, could be used to derive initial uncertainty statistics and reduce the initial ensemble spread.

  Finally, in our synthetic applications, we have not accounted for all potential sources of uncertainty which are, for example:

  – the ice flow law: the ice viscosity depends on the englacial temperature which itself is function of the ice sheet history
30   and the boundary forcing, including the geothermal heat flux (e.g. Van Liefferinge and Pattyn, 2013). Several other processes also affect the ice viscosity, including damage and strain-induced mechanical anisotropy (e.g. Pimienta et al., 1987; Schulson and Duval, 2009; Borstad et al., 2013). For the stress exponent, if the values $n = 3$ is used by most models, published values ranges between 1 and 5 (e.g. Gillet-Chaulet et al., 2011).

– The friction law: more and more direct or indirect evidences show that the friction under fast ice streams is at least partially controlled by the presence of sediments leading to a Coulomb type friction law (e.g. Tulaczyk et al., 2000; Murray, 1997; Joughin et al., 2010; Gillet-Chaulet et al., 2016). For hard beds, the development of subglacial cavities also implies deviations for the classical Weertman friction law (Schoof, 2005; Gagliardini et al., 2007).

– The density: the firn layer is not accounted for in most models, however its depth and density affect the floatation condition and thus the GL position (e.g. Griggs and Bamber, 2011). Direclty assimilating the GL position, using *e.g.* the moving mesh approach develop by Bonan et al. (2017), would certainly be beneficial in realistic applications to reduce the discrepancy between the modelled and observed GL (Goldberg et al., 2015).

– The external forcings from the atmosphere and the ocean: increasing mass loss rates from the ice sheets, in a large

portion, can be attributed to a response to oceanic forcing, but multiple challenges remain for a proper assessment of their magnitude (Joughin et al., 2012).

Realistic simulations with ice flow models cover a wide range of spatial and temporal scales, and the relative importance of these uncertainties as well as their representation in the models will certainly have to be evaluated partly in a case by case basis, requiring to develop robust framework for a variety of applications.

**6   Conclusions**

Developing model initialisation strategies that properly reproduce the ice-sheets dynamical mass losses observed over the last decades, requires to develop transient assimilation frameworks that are able to account for the growing availability of dense time series, especially from space observations. Here, we presented a synthetic twin experiment demonstrating the possibility to calibrate a marine ice model using an ensemble Kalman Filter which requires less numerical developments than variational

methods.

Using resolutions and noise levels consistent with current observing systems, good performances are obtained to recover both the basal friction and basal topography with an ensemble of at least 50 members. Localisation and inflation have been tuned manually, however the results are consistent over relatively wide ranges. Future studies should investigate how these values can be transposed to realistic applications. Nevertheless, there is an abundant and growing literature in other geophysical fields

to overcome problems that we might be facing in future studies.

Once the GL enters an unstable region, retreat rates largely depends on the basal conditions, thus using DA to reduce the associated uncertainties largely increases the skill of the model to predict rates and magnitude of GL retreat for time scales relevant for sea level rise projections. In our simplified application, the assimilation of the surface observations was sufficient to capture the GL migration during the assimilation window, without explicitely assimilating the observed position. However,

for the GL to enter a irreversible retreat, the thickness must reach a tipping point, *i.e.* the thickness at the GL must reach floatation. This can seriously impact the predictability of the system as, for small perturbations, remaining uncertainties on the basal conditions can lead to an uncertainty on the residence time of the GL on stabilisation points, that can be similar to the

simulation timescale. However, if the assimilation is pursued up to a time when the glacier is engaged in the unstable retreat, all the members exhibit the instability and the spread of centenial-scale model projections, in terms of volume and grounding line position, is largely reduced.

Finally, we have discussed the main challenges to tackle before generalizing transient DA in ice-sheet modelling. This includes a better assessment of the uncertainties in the model and in the observations used for the background and for the assimilation.

*Code availability.* Elmer/Ice code is publicly available through GitHub (https://github.com/ElmerCSC/elmerfem, Gagliardini et al. (2013)). PDAF is distributed under the GNU General Public License, version 3, and is available at http://pdaf.awi.de.

## Appendix A: Notations

**Table A1.** Notations and values used in this study associated with the ice flow model

| | | |
|---|---|---|
| Prognostic variables: | | |
| $H = z_s - z_b$ | m | Thickness |
| $z_s$ | m | top surface elevation |
| $z_b$ | m | bottom surface elevation |
| Diagnostic variable: | | |
| $u$ | $\mathrm{m\,a^{-1}}$ | horizontal velocity |
| Parameters: | | |
| $a_b = 0.0$ | $\mathrm{m\,a^{-1}}$ | basal melting |
| $a_s = 0.5$ | $\mathrm{m\,a^{-1}}$ | surface accumulation |
| $b$ | m | bed elevation |
| $B = 0.4$ | $\mathrm{Mpa\,a^{1/3}}$ | ice rigidity |
| $C$ | m | basal friction coefficient |
| $m = 1/3$ | | friction law exponent |
| $n = 3$ | | Glen's creep exponent |
| $\rho_i = 900$ | $\mathrm{kg\,m^3}$ | ice density |
| $\rho_w = 1000$ | $\mathrm{kg\,m^3}$ | sea water density |
| Numerical parameters: | | |
| $dt = 5\,10^{-3}$ | a | model time step |
| $dx = 200$ | m | mesh resolution |

**Table A2.** Notations and values used in this study associated with the ensemble filter

| | | |
|---|---|---|
| Variables: | | |
| $\boldsymbol{x} = (z_s, b, C)$ | | state vector |
| $\mathbf{P}$ | | covariance matrix |
| Stabilisation parameters: | | |
| $r$ | m | localisation radius |
| $\rho$ | | forgetting factor |
| Sizes: | | |
| $N_e$ | | ensemble size |
| $N_x$ | | state vector size |
| $N_y$ | | observation vector size |
| Others: | | |
| $\Delta t = 1$ | a | time interval between two analyses |

*Competing interests.* The authors declare that they have no competing interests.

*Acknowledgements.* This work was funded by the French National Research Agency (ANR) through the TROIS-AS (ANR-15-CE01-0005-01). The author thanks G. Durand, O. Gagliardini and J. Mouginot for valuable comments on the first drafts of the manuscript.

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

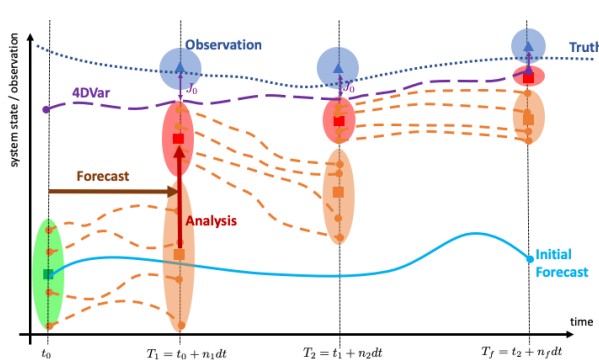

**Figure 1.** Principle of data assimilation (Adapted from Carrassi et al. (2018)). Having a physical model able to forecast the evolution of a system from time $t = t_0$ to time $t = T_f$ (cyan curve), the aim of DA is to use available observations (blue triangles) to correct the model projections and get closer to the (unknown) truth (dotted line). In EnKFs, the initial system state and its uncertainty (green square and ellipsoid) is represented by $N_e$ members. The members are propagated forward in time during $n_1$ model time steps $dt$ to $t = T_1$ where observations are available (Forecast phase, orange dashed lines). At $T = t_1$ the analysis uses the observations and their uncertainty (blue triangle and ellipsoid) to produce a new system state that is closer to the observations and with a lower uncertainty (red square and ellipsoid). A new forecast is issued from the analysed state and this procedure is repeated until the end of the assimilation window at $t = T_f$. The model state should get closer to the truth and with lower uncertainty as more observations are assimilated. Time dependent variational methods (4D-Var) iterate over the assimilation window to find the trajectory that minimizes the misfit ($J_0$) between the model and all observations available from $t_0$ to $T_f$ (violet curve). For linear dynamics, Gaussian errors and infinite ensemble sizes, the states produced at the end of the assimilation window by the two methods should be equivalent (Li and Navon, 2001).

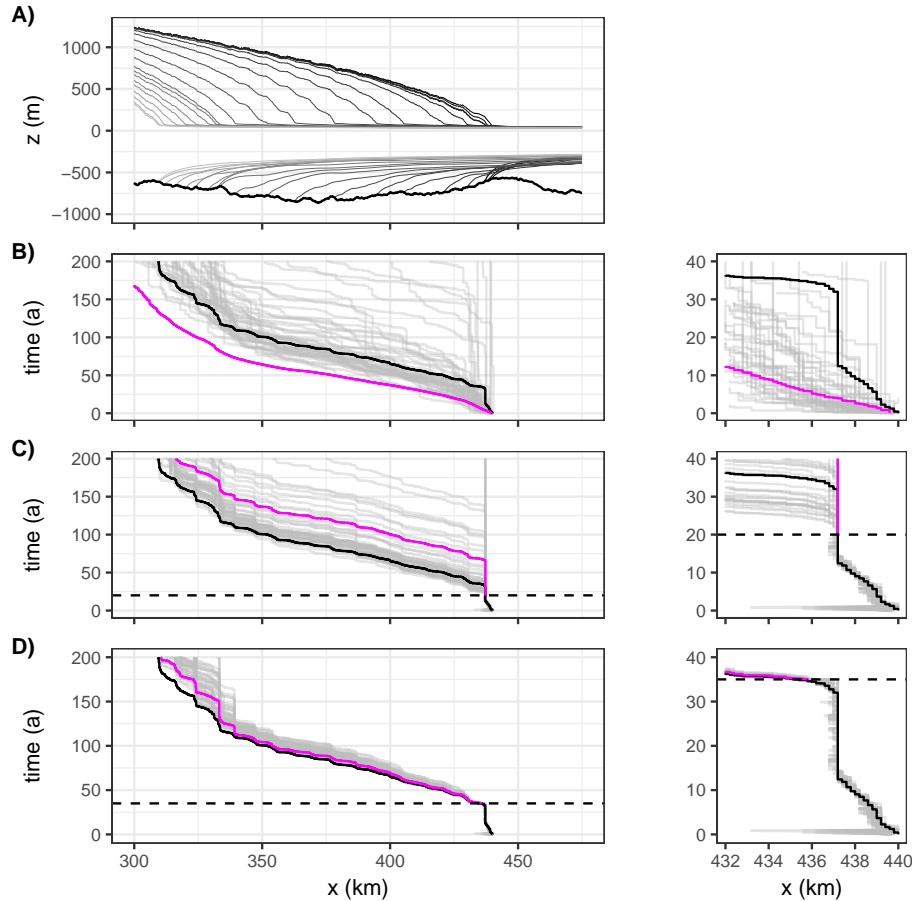

**Figure 2.** (A) Reference ice sheet topography every 10 years from $t = 0$ to $t = 200\,\mathrm{a}$ (black to grey). (B-D) GL position as a function of the simulation time for the reference (black line), for the ensemble (grey lines), and for the deterministic forecast (magenta line) (B) without assimilation, (C) with assimilation up to $t = 20\,\mathrm{a}$ and (D) with assimilation up to $t = 35\,\mathrm{a}$. The right column show a zoom on the first 40 years. In C-D), the horizontal dashed line show the end of the assimilation window.

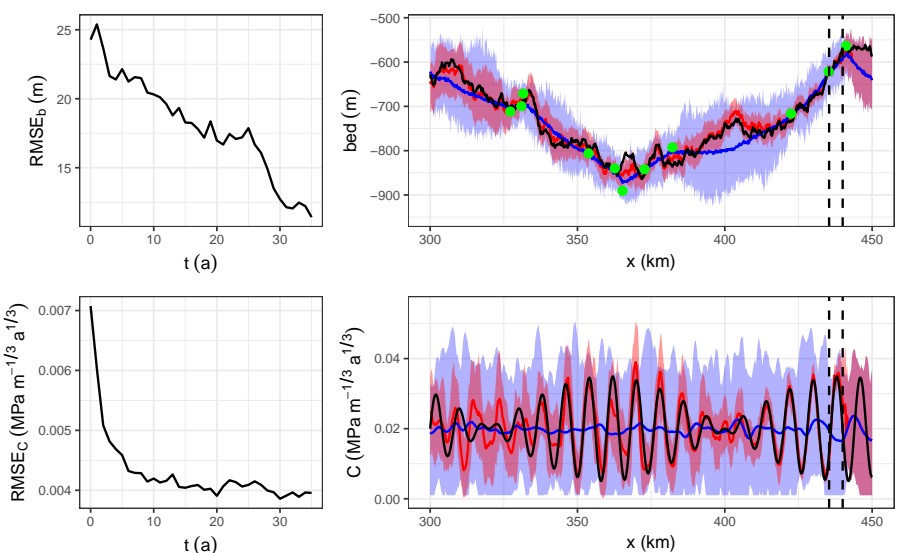

**Figure 3.** (left) RMSE between the reference and the analysed ensemble mean for the bed and friction coefficient. (Right) Bed and friction coefficient, the reference is shown in black, the synthetic bed measurements in the top panel are shown as green dots, the ensemble mean before assimilation is in blue and at $t = 35\,\mathrm{a}$ in red. The shading shows the ensemble spread between the minimum and maximum values, before assimilation (blue) and at $t = 35\,\mathrm{a}$ (red). The dashed vertical lines show the GL position at $t = 0$ and $t = 35\,\mathrm{a}$.

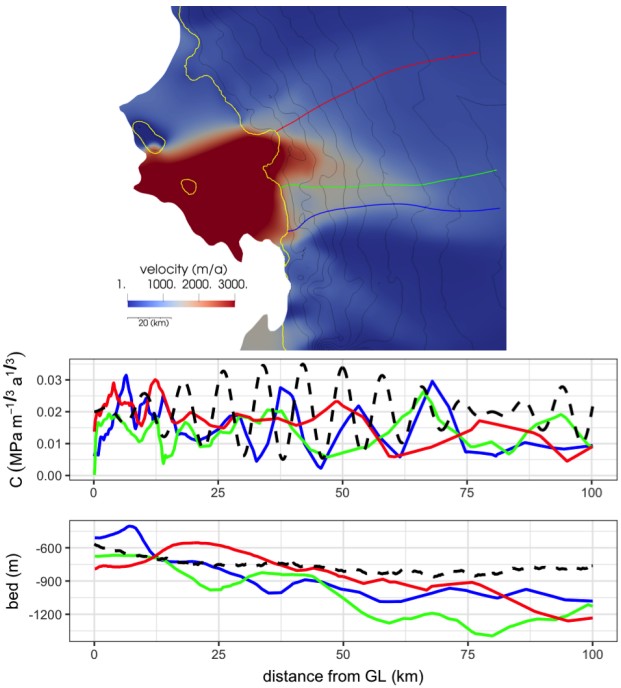

**Figure 4.** Thwaites Glacier (Antarctica). Model results from Brondex et al. (2018): model velocities (top) and friction coefficient $C$ and bed elevation $b$ extracted along three streamlines (same color code). Synthetic values used in this study are shown with black dashed lines. Note that the mesh resolution varies from $\backsim$ 200m close to the GL, shown in yellow in the top panel, to $\backsim$ 10km at the upstream end of the streamlines.

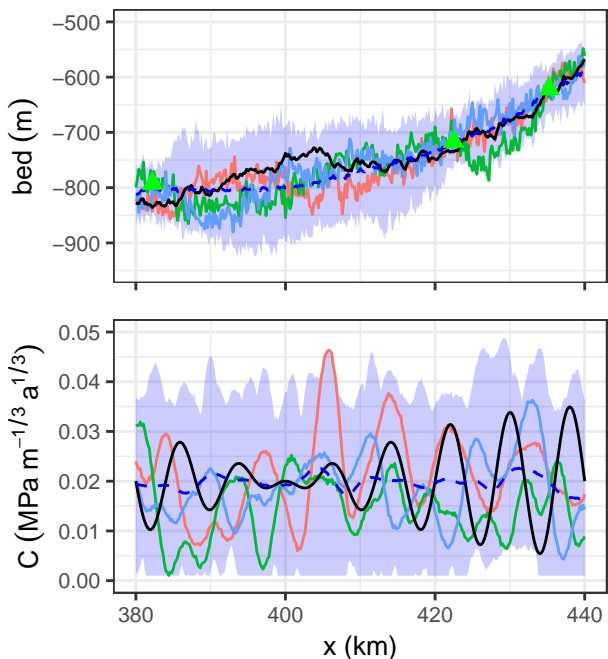

**Figure 5.** Initial ensemble for the bed and friction coefficient, the reference is shown in black, the synthetic bed measurements are shown as green triangles in the top panel, the ensemble mean is the dashed blue curve and the shading shows the ensemble spread. Coloured solid lines show the first 3 members.

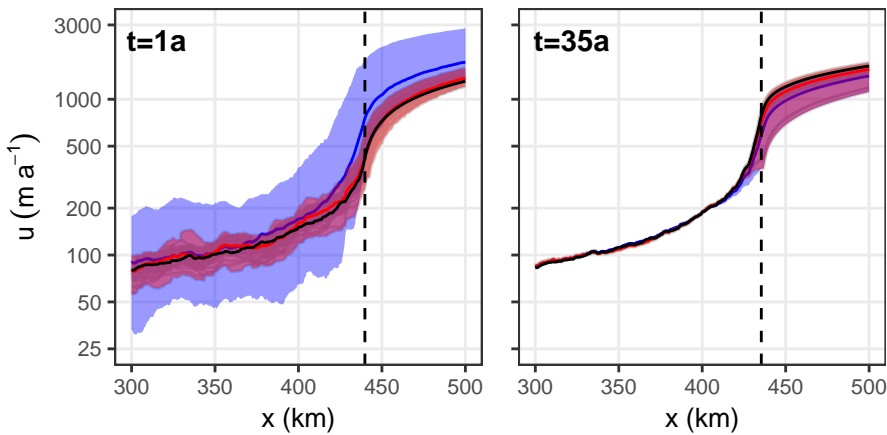

**Figure 6.** Velocity, $u$, at $t = 1$a and $t = 35$a. The reference is in black, the ensemble mean before and after the analysis is in blue and red, respectively. The shading shows the ensemble spread between the minimum and maximum. The dashed vertical black indicates grounding line position.

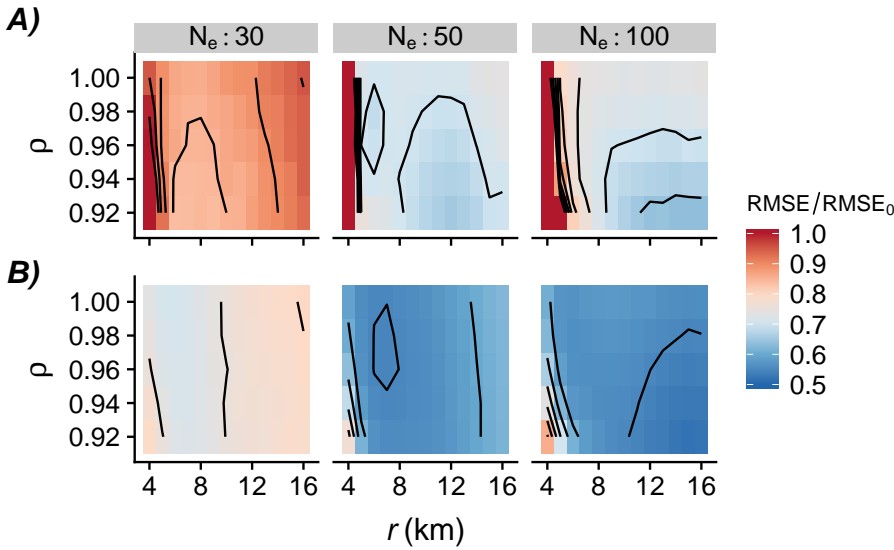

**Figure 7.** RMSE at $t = 20a$, relative to the initial (before assimilation) RMSE for the 50-member ensemble as a function of the forgetting factor $\rho$ and the localisation radius $r$ for different ensemble sizes $N_e$. A) for the bed and B) for the friction coefficient. Black lines as isovalues spaced by 5%.

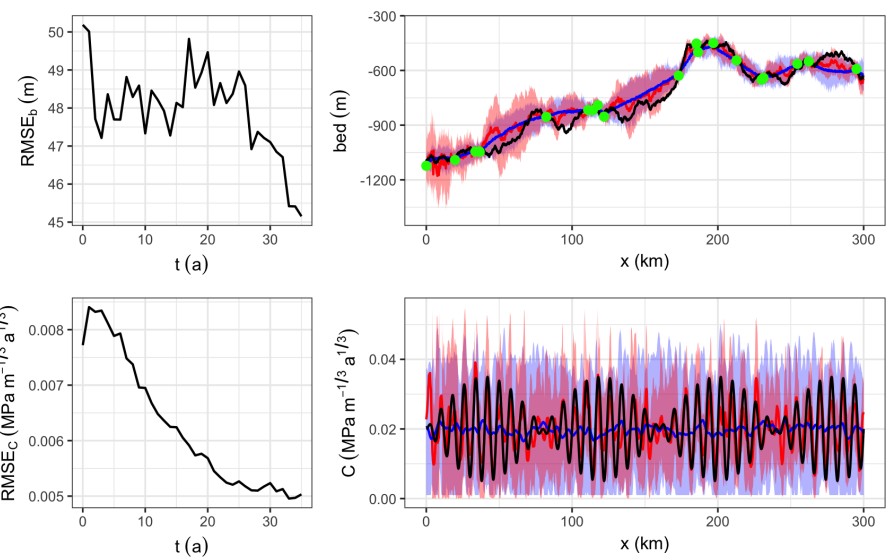

**Figure 8.** Same as Fig. 3 but with the RMSE computed for $x \in [0, 300]$ km.

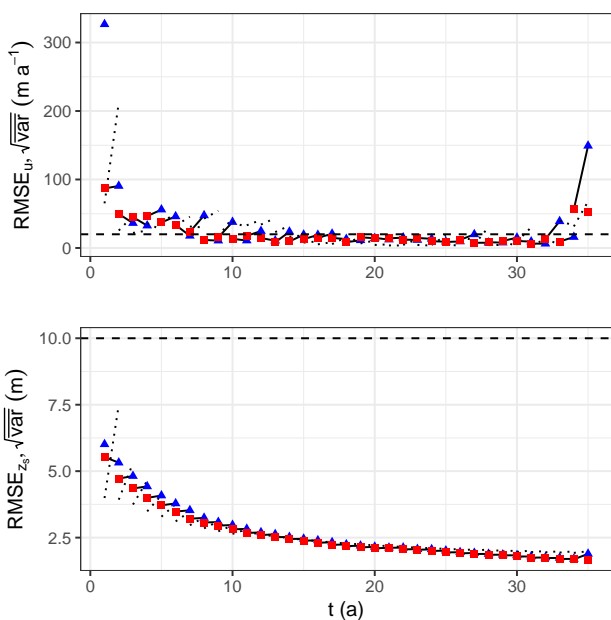

**Figure 9.** RMSE (solid lines) and square root of the averaged ensemble variance (dashed lines) during the assimilation window for (top) the velocity, $u$, and (bottom) the free surface, $z_s$. Each year, the blue triangle and the red square are the RMSEs before and after the analysis, respectively. Each segment represent a 1-year forecast step.

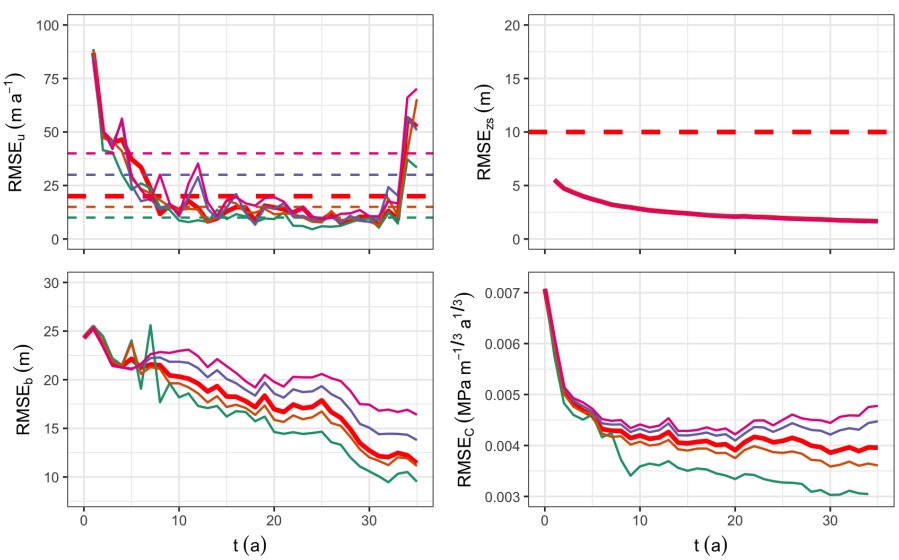

**Figure 10.** Sensitivity to the surface velocities observation error $\sigma_u^{obs}$ : RMSEs after each analysis, computed only for $x \geq 300km$ for $b$ and $C$.. The thick red lines correspond to the results with $\sigma_u^{obs} = 20$ ma$^{-1}$ and $\sigma_{z_s^{obs}} = 10$ m shown in Figs. 3 and 9. The horizontal dashed lines correspond to the observation errors $\sigma_u^{obs}$ and the results are presented with solid lines using the same color code. $\sigma_{z_s^{obs}} = 10$ m for all the experiments.

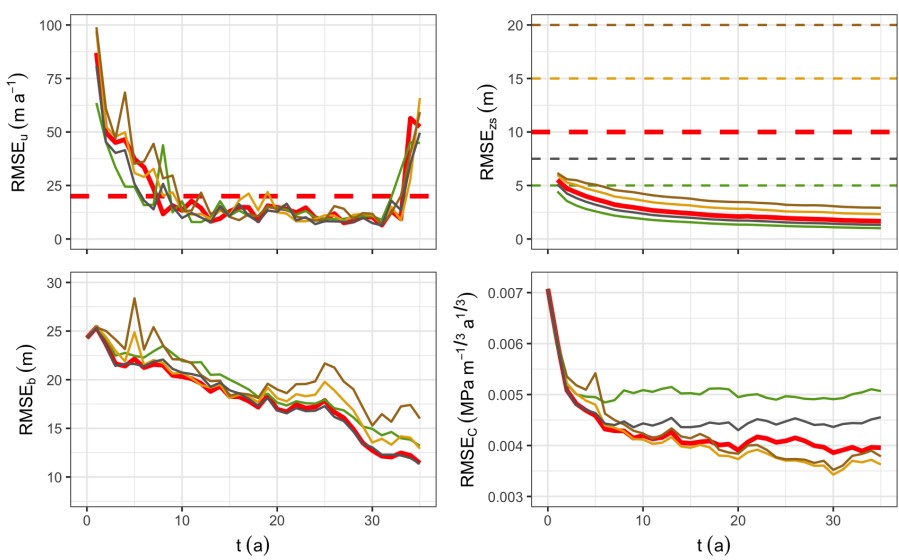

**Figure 11.** Sensitivity to the surface elevation observation error $\sigma_{z_s^{obs}}$: RMSEs after each analysis, computed only for $x \geq 300km$ for $b$ and $C$. The thick red lines correspond to the results with $\sigma_u^{obs} = 20$ ma$^{-1}$ and $\sigma_{z_s^{obs}} = 10$ m shown in Figs. 3 and 9. The horizontal dashed lines correspond to the observation errors $\sigma_{z_s^{obs}}$ and the results are presented with solid lines using the same color code. $\sigma_u^{obs} = 20$ ma$^{-1}$ for all the experiments.

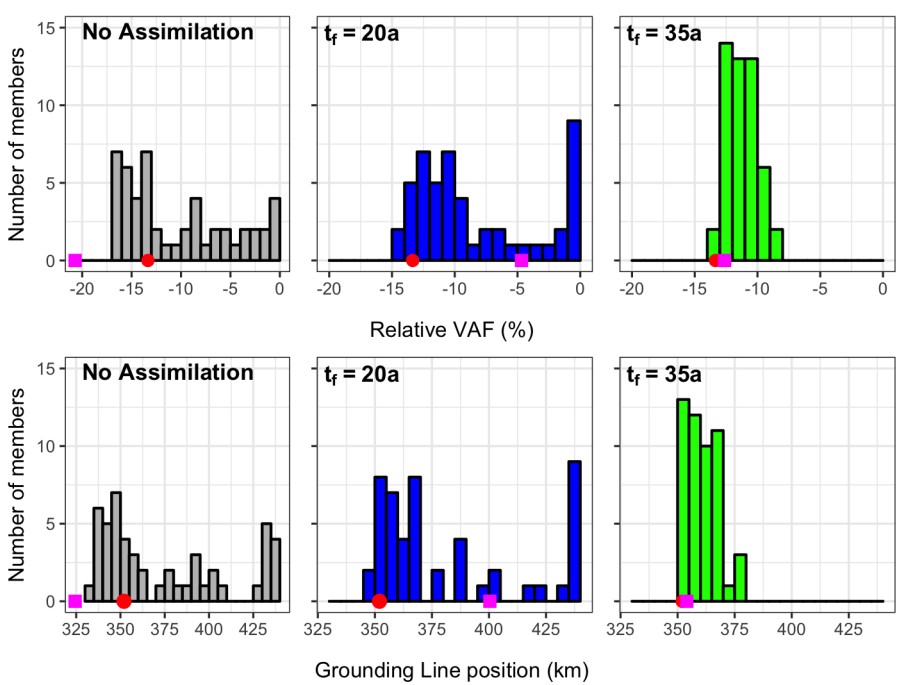

**Figure 12.** Ensemble forecast at $t = 100$ a: (top) relative change of volume above floatation (VAF) and (bottom) GL position with (left) no assimilation, (center) assimilation up to $t = 20$ a and (right) assimilation up to $t = 35$ a. The red circle correspond to the reference run and the magenta square to the deterministic forecast.