# Peer review of "Assimilation of surface observations in a transient marine ice sheet model using an ensemble Kalman filter"

_The Cryosphere, 2019_

## Referee Comment (RC1) · Dan Goldberg (Referee) · 28 Jun 2019

This study is one of the first to apply Ensemble Kalman Filtering methods to an ice-sheet model with a nontrivial stress balance (attempts have been made with Shallow-ice models). Such methods, rather than using deterministic means to optimise a cost/misfit function in order to infer hidden properties from error-prone observations, essentially generate an ensemble meant to encompass a probability distribution, and repeatedly apply model dynamics and bayesian inference on this ensemble in order to refine the statistical properties of an unknown state and parameter set. The paper implements a variant of the EnKF known as the Ensemble Subspace Transform

[Figure]

Kalman Filter, which is simply a particular choice and one which is meant to avoid costly computation of an ensemble covariance (which i do question – please see specific comments).

The methodology presented is really just one step in a long long road toward operational ice-sheet forecasting (compare with over a half-century of development in numerical weather prediction) but an important one – especially when considering there is at least one person with a question about filtering methods every time a talk is presented in state and parameter estimation for ice sheets (as the authors have pointed out, there are already others working toward the ice-sheet version of the other main tool of weather prediction, 4Dvar). Thus i feel it is a worthwhile study which should be worthy of publication as a methodological investigation (and the authors frame it as such, at least in the conclusions section. However I do think the manuscript needs work before this can happen.

On the basis of the extent of the comments below I choose "major revisions" – but there is no formal definition of what this means, and the editor may choose to ignore this classification. I am not suggesting modification of the algorithm and/or results, simply clarity of text.

GENERAL COMMENTS

For one thing upon reading I had significant detail understanding what was done. There were a number of points on which i felt clarity was needed, and most of these are addressed below in line-by-line comments so I will not list them here. Note that these specific questions compose the bulk of my review – and this is because without having a better idea of what was actually implemented, it is difficult to critique the results further!

However something I will state in the general comments is that despite similarities, ice-sheet models are distinct from e.g. atmospheric models in that the unknown parameters most sought cannot generally be observed directly, in contrast to models in which

the initial conditions in a forecast/analysis cycle represent the parameters of greatest interest. This is exemplified by the fact that the "state" vector contains non-dynamic variables (friction and bed elevation) and the fact that the observation operator, rather than being a simple averaging or restriction, encapsulates a fully nonlinear solve of an elliptic partial differential equation. I think this is something that should be made very clear to members of the climate and meteorological community who read this work.

One overall comment is regarding the distribution of the ensemble. As I understand it, even if the initial ensemble is evenly distributed given the prior, it is difficult to know a priori whether the projection of the ensemble will represent a favorable distribution of the projected space. That is, what if the forecast "clusters", underrepresenting important regions of state space? As I ask below, it is unclear whether there is a "reinitialisation" of the ensemble at every step. Clearly this topic has already been considered in the NWP literature, for example Song et al (2013) makes use of a time-dependent adjoint (a tool the authors state the work here is meant to circumvent the need for) in order to generate a more representative ensemble.

I also question whether the "toy problem" proposed by the authors truly tests all of the difficulties a filtering approach might encounter. I bring this up in more detail below but some aspects of the approach seem to hang on the "locality" of the problem (a technique called "localisation" is employed to ignore long-distance correlations of the state). I wonder if this only works because the problem is one-dimensional with no buttressing involved, so essentially to a strong degree (though not completely) the velocities depend locally on basal friction and geometry? Would this still be a good approach in a 2D domain with an expansive embayed ice shelf (such as the Ross or FRIS), or very weak basal traction over a large part of the domain (such as Pine Island)?

The methodology essentially uses a whole "family" of geometries and velocities to infer hidden parameters of the system. This somewhat bears similarity to a differnet paper led by the author, "Assimilation of surface velocities acquired between 1996 and 2010

to constrain the form of the basal friction law under Pine Island Glacier" – aside from the statistical formality, and the introduction of consistency between these geometries by way of the continuity equation (which is actually not so consistent if the analysis updates do not conserve mass!!!) – I wonder if the author would consider comparing and contrasting these approaches.

Finally, i point out that, despite the divide between filtering and adjoint-based methods, there is a growing sentiment in NWP to take what is "best" from the various approaches and form more hybrid schemes (for instance the Song paper referenced above, see also Kalnay 2010). Therefore I urge the author to reflect on such innovations and how they might be useful in further developments for filtering of ice-sheet models.

Line by Line comments:

p2.l3: would be good to state this is a point when only resolving 1 horizontal dimension

p3.10 variational

p3.13 "use of linearised or adjoint models" this assumes a trivial mapping from model vars to observations – see my comments below.

p3.15 rewrited?

p3.35 – would be good to explain as soon as possible i.e. here what you mean by a twin experiment, or give a reference, as this is jargony

p 5 thru eq (10): this is a well written explanation of the EnKF. However I have a few questions which might be due to my lack of familiarity with filtering methods, but I think this might be true of many readers of this paper. This is also important as, though it is not the algorithm used, the one used is far more complex so this is a chance to explain your methods to the reader.

(a) is $P^f = P_k$?

(b) you do not say how the individual ensemble members ($x_i^{k,a}$) arise/are updated,

only the state vector (which looks like the mean of the analysed ensemble)?

(c) is the posterior/analysis covariance used at all in subsequent time/filtering steps, as from eq 7 the covariance is always formed from the present ensemble – so i am struggling to grasp what is done in the algorithm in a multi-time step (k>1) framework.

(d) For each new forecast/analysis cycle, is the ensemble generated anew from the analysis-generate ensemble statistics?

(e) the formula given assumes normality of the ensemble does it not?

P5 eq(8): $M_k$ is trivially the identity on the time-invariant components of $x$, i.e. $b$ and $C$, correct?

p6.4. I am struggling to see why $\hat{P}^f$ need be formed, as it is a tensor product of X with itself (subject to (a) above). For instance, the last term in 9(a) is written

$X (X^T H^T) ((H X) (X^T H^T) + R)^{-1}$ dimension

so the largest matrix that need be formed is HX, and no matrix of (Nx x Nx) need be formed. Perhaps I do not understand where and how $\hat{P}^a$ is actually used however.

P6.5 I don't feel the concept of "error subspace" is ever suitably explained as i read the paper still wondering about this. \Omega as defined in eq 11 simply seems to be a "mixing" matrix that slightly changes the ensemble members – how is this an "error subspace"? (Assumiing that X \in R^{Nx x Ne} – i take it this is the case for eq 11 to make sense...)

Eqs 11-16: in contrast to the discussion of the EnKF this is very nonintuitive. You state (P6 line 7) that you approximate the covariance matrix by a low-rank matrix, which seems intuitive, but where is the equation describing this low-rank approximation and how is it done? (for what it is worth, low-rank approximations of covariance generally involve eigenvalue decompositions to retain the leading order covariance structure, but i do not see this here...)

[Figure]
**TCD**

Interactive
comment

P6.23-25: can you give a more intuitive description of inflation? Why do you need it and what does it achieve? As it is I am not even sure if inflation corresponds to lower or higher rho.

P7, first paragraph. Im sorry but I am struggling to follow this paragraph. For instance, how does the non-linear observation operator applied to x_i lead to the product HXˆf? i imagine they are related, as H is a linearisation of $H$ (which, by the way, clashes with the symbol for ice thickness) but this is not explained.

P7.8. This assumption, i imagine, is valid in many NWP settings given the hyperbolicity of the equations. Are you confident it is a good approach for marine ice-sheet modelling?

P8.26. I was surprised by your suggestion that annual DEMs would be available over a multiple decadal period, as i do not know of such products for antarctica. The best i have seen is decadal or semidecadal with MUCH lower spatial resolution (e.g. Konrad et al 2017, GRL). Having skimmed the ArcticDEM website i do see mention of the spatial resolution, but not the temporal. Unless you can argue that such spatiotemporal resolution is reasonable and available, i suggest caveating this discussion by saying it is an idealised experiment and this is the type of spatiotemporal resolution to which the community should aspire.

P9.6: prognostic ice sheet models generally step forward the ice thickness, not surface elevation (as shown in your eq 4). In the analysis step you are updating z_s. Is there a simple mapping from your model state X to thickness?

P9.6: as mentioned in the previous comment you are updating z_s in each analysis step, which i am inferring then maps on to an update in thickness (tell me if i am wrong). Is this update at all volume conserving? If not should this be a concern?

P9.21-28: Lots of jargony language in this paragraph, likely not to be understood by the target audience. What is a sill and a nugget? You talk of the prediction obtained by
kriging – is this something you have calculated? Is there any way to evaluate whether the ensemble does converge to it? Is this a way of evaluating whether the ensemble is large enough?

Section 4.1: Upon reading this, I realised that (a) i am unsure what time step you used, and (b) more importantly, whether each time step is a forecast/analysis step, as M_k in eq 8 could easily encompass multiple time steps – is this the case?

P10.15: again, these factors seem very important, and as mentioned above not overly well explained.

Section 4.2 – section headings should be capitalised

P12.25: Two comments about this paragraph: (a) Code that is continuously being up-dated and new algorithms developed might be an issue for *analytically* derived adjoint models, but not as much for automatic differentiation, which is specifically designed to generate new adjoint code when the "primal" code is changed; and (b) I return to the my confusion over the first paragraph on P7. Your observation matrix H contains, at the very least, a linearisation of the stress balance equation mapping geometry and basal friction onto velocity. It is not clear how you are finding this operator if not through some sort of forward model linearisation.

References:

Song et al, 2013: An adjointbased adaptive ensemble Kalman filter. Mon. Wea. Rev., 141, 3343–3359, https://doi.org/10.1175/MWR-D-12-00244.1.

Kalnay, E. (2010), Ensemble Kalman Filter: Current status and potential, in Data Assimilation: Making Sense of Observations, edited by W. Lahoz, B. Khattatov, and R. Menard, 24 pp., Springer, New York

---

## Referee Comment (RC2) · Anonymous Referee #2 · 16 Jul 2019

First of all, I would like to apologise to the author and the editor for providing my review so late. I consider this paper treats an important subject in an innovative manner and, as such, deserves a careful review. I hope you will find my review insightful.

**OVERVIEW**

The paper aims to adapt an Ensemble Kalman Filter (EnKF) to estimate jointly the surface elevation, the bedrock topography and the basal friction coefficient in the case of a flowline marine ice sheet. Time-dependant ensemble data assimilation (DA) approaches are relatively new for ice sheet initialisation. The paper focuses on the case of a grounding line retreat for unstable glaciers, which is a hot topic in ice sheet modelling and climate change. It also studies the influence of DA on forecasts of grounding line retreat.

**GENERAL COMMENTS**

The paper targets an important and timely question: how to initialise and estimate basal parameters to forecast the evolution of marine ice sheets and glaciers especially in the case of an unstable retreat? and proposes to use an EnKF (here an ESTKF) in this context. EnKFs have shown how efficient they can be in a wide range of applications (not exclusively meteorology and oceanography but also hydrology, crop modelling, oil extraction, pollutant dispersion, . . .), are a good alternative to adjoint-based methods and are the basis of hybrid methods that are now popular in DA (see e.g. Bannister, 2017). The paper shows clearly that EnKFs are a good toolbox for DA in glaciology. The experiment shows clearly how beneficial this approach could be with adequate figures and a very insightful analysis. Overall I am convinced that the paper in its final form will be a very good introduction of ensemble DA methods in ice sheet modelling.

Nevertheless, there are few points that prevent me to publish the paper as is. I list them below:

- I have some reservation about the experimental setup mainly on how the reference run is designed and on how the basal friction coefficient $C$ is estimated.
  - About the reference run: The long-term objective of the study is to forecast accurately grounding line retreat for Antarctic hemispheric glaciers in the context of global change. However, in the reference run of the experiment, the grounding line retreat is triggered by the abrupt change of ice rigidity

$B$. I wonder if the simulated grounding line retreat is "realistic" compared to one triggered by climate change. Also I have the same question about the sinusoidal basal friction parameter $C$. How realistic is it compared to real cases? I know we are in flowline cases using SSA equations. But it would strengthen the paper if the author could reflect on that subject in the section 3.1. Part of my comment may be due to my lack of knowledge in glaciology, so please accept my apologies if my comment is irrelevant.

– About the assimilation setup: The basal friction coefficient $C$ must be positive. To ensure such thing, you use the following change of variable $C = \alpha^2$. But by doing so, you do not ensure the unicity of the estimation as $\alpha$ and $-\alpha$ would lead to the same $C$. Also I thought that the $C$ parameter could be of different order of magnitude. To counteract those potential issues, Bonan et al. (2014) chose the following change of variable $C = 10^\alpha$ ensuring the positivity of $C$, the unicity of the change of variables and mimicking behaviours with different order of magnitude. Could you explain why the change of variable you used is more appropriate in your context?

• The paper is full of interesting results but sometimes they deserve a more thorough analysis.

– The assimilation window is between 1 yr and 35 yr (you run forecasts from analysed states at $t = 20$ yr and $t = 35$ yr). But the grounding line is almost steady between $t = 13$ yr and $t = 32$ yr. We also see that after the first 10 years of assimilation, RMSD for $C$ remains stable. I wonder if those two points are correlated meaning is it easier or more difficult to estimate $C$ in the case of a retreating grounding line (hence a more dynamic ice sheet) or an almost steady state? As the behaviour of grounding line is different from one marine glacier to another, it would be beneficial to the community if you could push your study in that direction in the revised version of the

manuscript (for example continuing DA after $t = 35$ yr for the next ten years and study what happens).

– Figure 5 shows interesting results about the performance of ESTKF with inflation and localisation and varying sizes of the ensemble. Bonan et al. (2014) has performed the same kind of studies for grounded ice sheet using an ETKF. Do you obtain similar optimal parameters for inflation and localisation in your experiment or are they different from Bonan et al. (2014)? It would be interesting to reflect on how the physics of the model influences such parameters.

– You only show results after 300 km (nothing between 0 km and 300 km). Does it mean that what happens between 0 km and 300 km does not have an influence on the grounding line retreat? Does it mean DA is pointless in those areas (in that case, that would make DA more affordable as less grid points need to be treated)? If so, please state it more clearly and if not, provide more results for that area.

– I am worried by some of the results shown for ESTKF for small ensembles ($N_e = 30$) as RMSD reduction is only 20% for the bedrock topography and 30% for the basal friction coefficient. For 2D real cases (either using Full-Stokes or SSA), we need an ensemble approach working for small ensembles due to the cost of the experiment and 30 members might be too expensive. Could you reassure me and provide the reader that the approach would be beneficial even in 2D real cases?

– About the forecast experiments, Figure 1 shows clearly how beneficial an ensemble forecast can be compared to a deterministic forecast. It also shows that the distribution of the grounding line position is not Gaussian. Could you provide more information (maybe using histograms) on how grounding lines are distributed in the ensemble?

• While on average, the paper is perfectly readable. There are few parts that are

hardly accessible to the reader. As I think the target audience of this paper is the ice sheet community and more generally the glaciology community, the paper would benefit from a careful editing. I detail those parts in the Specific comments section.

Overall I consider the paper as a highly valuable addition to data assimilation for ice sheets. But it deserves mostly some rewriting. Therefore I recommend a minor revision as almost all the science is already here!

**SPECIFIC COMMENTS**

Overall I think it would be nice for the reader to have two tables summarising the various variables used in this paper, one for the ice flow model and one for data assimilation.

About Ensemble Kalman Filters:

- I feel localisation and inflation should be better explained in the paper (either in the introduction and in the DA section). I agree they are both used to counteracts the effects of undersampling. But the reader would benefit from having more information. Undersampling causes underestimated variances (counteracted by inflation) and spurious correlations (counteracted by localisation in the case of long-range spatial spurious correlations). Could you add few lines on the subject. Also few references are missing. For inflation:

  Anderson, J. L. and Anderson, S. L.: A Monte Carlo implementation of the nonlinear filtering problem to produce ensemble assimilations and forecasts, Mon. Weather Rev., 127, 2741–2758, doi: 10.1175/1520-0493(1999)127<2741%3AAMCIOT>2.0.CO%3B2, 1999.

For localisation, the first one is for local analysis (the one you use), the other two are for covariance localisation (the historical one)

Ott, E., Hunt, B. R., Szunyogh, I., Zimin, A. V., Kostelich, E. J., Corazza, M., Kalnay, E., Patil, D. J., and Yorke, A.: A local ensemble Kalman filter for atmospheric data assimilation, Tellus A, 56, 415–428, doi: 10.1111/j.1600-0870.2004.00076.x, 2004.

Hamill, T. M., Whitaker, J. S., and Snyder, C.: Distance-dependent filtering of background error covariance estimates in an ensemble Kalman filter, Mon. Weather Rev., 129, 2776–2790, doi: 10.1175/1520-0493(2001)129<2776%3ADDFOBE>2.0.CO%3B2, 2001.

Houtekamer, P. L. and Mitchell, H. L.: A sequential ensemble Kalman filter for atmospheric data assimilation, Mon. Weather Rev., 129, 123–137, doi: 10.1175/1520-0493(2001)129<0123%3AASEKFF>2.0.CO%3B2, 2001.

- Also about inflation, the term "forgetting factor" introduced by Pham et al. (1996) is, unfortunately, very uncommon in the EnKF community. Could you state somewhere that this is just the inverse of the traditional inflation parameter known widely in the EnKF community?

- p. 6, l. 23: There is no unicity of the symmetric square root matrix $\mathbf{C}$. It is known that the choice of $\mathbf{C}$ can have a significant impact on results (see e.g. Livings et al., 2008). Could you detail how $\mathbf{C}$ is calculated in PDAF?

Livings, D. M., Dance, S. L., and Nichols, N. K.: Unbiased ensemble square root filters, Physica D, 237, 1021–1028, doi: 10.1016/j.physd.2008.01.005, 2008.

- p. 7, first paragraph on how to use the ESTKF with a nonlinear observation operator. It is a very good point you raise especially in the case of assimilating surface ice velocities (highly nonlinear observation operator). However, I find it difficult to see where the nonlinearity of the observation operator intervenes.

Could you rewrite the whole section 2.2 and consider directly the case when the observation operator is nonlinear? That would avoid confusion for readers.

About the experiment:

- p. 8, l. 1-2: About the roughness signal $b_r$ ,could you detail, in annex for example, how do you simulate the roughness (which equation?) because I do not know this approach.

- p. 9, l. 21-22: you mention the term "*variogram*" which is not well known for readers. Could you provide more details how do you generate the ensemble of initial bedrock topographies, in annex for example?

- p. 9, l. 30-31: same comment as previous.

About the discussion:

- p. 12, l. 25-29: you seem to oppose ensemble and variational methods, but more and more, the tendency is to develop hybrid methods as detailed in Bannister (2017) and Vetra-Carvalho et al. (2018). The main tendency is to use variational approaches in which the adjoint is replaced by ensembles making those adjoint-free approaches. Could you modify your paragraph to reflect this tendency?

- p. 13, l. 4-13: There is a now long range of DA literature on how to estimate model bias. One good reference is the following:

  Dee, D. P. Bias and data assimilation, Q. J. Roy. Meteor. Soc., 131, 3323–3343, doi: 10.1256/qj.05.137, 2005.

  Could you reflect on that possibility in your discussion?

[Figure]

- p. 13, l. 14-20: Same comment as before on estimating observation error covariances matrices. A good review paper:

Tandeo, P., Ailliot, P., Bocquet, M., Carrassi, A., Miyoshi, T., Pulido, M. and Zhen, Y.: Joint Estimation of Model and Observation Error Covariance Matrices in Data Assimilation: a Review. Mon. Weather Rev., submitted, available at: https://arxiv.org/abs/1807.11221v2, 2018.

Most approaches are based on Desroziers diagnostics, see:

Desroziers, G., Berre, L., Chapnik, B. and Poli, P.: Diagnosis of observation, background and analysis-error statistics in observation space. Q. J. Roy. Meteor. Soc., 131, 3385–3396, doi: 10.1256/qj.05.108, 2005.

**MINOR COMMENTS AND TYPOS**

- p. 1, l. 5: "*starting FROM this initial state . . .*"

- p. 3, l. 15: "*the Kalman filter analysis is REWRITTEN and . . .*"

- p. 3, l. 16: The references you mention are all about deterministic versions of the EnKF (Pham et al., 1998, SEIK filter; Bishop et al., 2001, ETKF filter: Nerger et al., 2012, ESTKF filter). But not every EnKF has a deterministic analysis, the stochastic EnKF has also been an important part of EnKF history. Could you add the following references to make your point broader?

Burgers, G., van Leeuwen, P. J., and Evensen, G. Analysis scheme in the ensemble Kalman filter, Mon. Weather Rev., 126, 1719–1724, doi: 10.1175/1520-0493(1998)126<1719:ASITEK>2.0.CO;2, 1998.

Houtekamer, P. L., and Mitchell, H. L. Data assimilation using an ensemble Kalman filter technique, Mon. Weather Rev., 126, 796-811, doi: 10.1175/1520-0493(1998)126<0796%3ADAUAEK>2.0.CO%3B2, 1998.

- p. 3, l. 23: "*However the many applications in meteorology and oceanography show . . .*" While EnKFs have been primarily developed for those two applications, it has been successfully used in a wide range of applications, from hydrology, to crop modelling and oil extraction. Could you rephrase the sentence to show the broad range of applications for EnKF including some that may be closer to glaciology? Maybe add other references too?

- p. 6, Eq. (13): Could you define $\bar{\mathbf{X}}^f$?

- p. 9, l. 13: "*the transient ASSIMILATION ON model projections*"

---

## Author Comment (AC1) · 28 Aug 2019

Dear Editor, dear reviewers,
I really want to thank the two reviewers for their comments that I found very complementary and helpful to better explain the aims and results of this paper.
Both reviewer agreed that this paper has the potential of providing a good introduction of ensemble DA methods for the ice sheet modelling community.
To introduce the sequential DA algorithm and illustrate the difference with variational DA, I have added a new figure 1 (Fig. 1), that provides a schematic representation of the results expected with both methods.
Following the suggestions from the reviewers, I have mostly rewritten section 2.2 where I describe the assimilation setup. There is an abundant literature in geophysics on DA methods and it is not always easy to understand all the subtleties and differences between different methods. I hope that I have been successful now in giving a self-contained and clear introduction to the method.
To show that the synthetic experiment is realistic I have added a new Figure 3 (Fig. 2) where I compare the values for $b$ and $C$ used in this study to those obtained by Brondex et al. (2018) for Thwaites Glacier in Antarctica.
For the interpretation of the results I have performed additional experiments where I study the DA performance in retrieving the basal conditions as a function of the noise level in the surface observations (Figs. 8 and 9).
You will find below my point by point answer to the reviewer comments, with the original comments in black and my answers in red

[Figure]

**Figure 1.** NEW FIGURE 1. Principle of data assimilation (Adapted from Carrassi et al. (2018)). Having a physical model able to forecast the evolution of a system from time $t = t_0$ to time $t = T_f$ (cyan curve), the aim of DA is to use available observations (blue triangles) to correct the model projections and get closer to the (unknown) truth (dotted line). In EnKFs, the initial system state and its uncertainty (green square and ellipsoid) is represented by $N_e$ members. The members are propagated forward in time during $n_1$ model time steps $dt$ to $t = T_1$ where observations are available (Forecast phase, orange dashed lines). At $T = t_1$ the analysis uses the observations and their uncertainty (blue triangle and ellipsoid) to produce a new system state that is closer to the observations and with a lower uncertainty (red square and ellipsoid). A new forecast is issued from the analysed state and this procedure is repeated until the end of the assimilation window at $t = T_f$. The model state should get closer to the truth and with lower uncertainty as more observations are assimilated. Time dependent variational methods (4D-Var) iterate over the assimilation window to find the trajectory that minimizes the misfit ($J_0$) between the model and all observations available from $t_0$ to $T_f$ (violet curve). For linear dynamics, Gaussian errors and infinite ensemble sizes, the states produced at the end of the assimilation window by the two methods should be equivalent (Li and Navon, 2001).

**REVIEWER 1, Dan Goldberg**

This study is one of the first to apply Ensemble Kalman Filtering methods to an ice-sheet model with a nontrivial stress balance (attempts have been made with Shallow- ice models). Such methods, rather than using deterministic means to optimise a cost/misfit function in order to infer hidden properties from error-prone observations, essentially generate an ensemble meant to encompass a probability distribution, and repeatedly apply model dynamics and bayesian inference on this ensemble in

order to refine the statistical properties of an unknown state and parameter set. The paper implements a variant of the EnKF known as the Ensemble Subspace Transform Kalman Filter, which is simply a particular choice and one which is meant to avoid costly computation of an ensemble covariance (which i do question – please see specific comments). The methodology presented is really just one step in a long long road toward operational ice-sheet forecasting (compare with over a half-century of development in numerical weather prediction) but an important one – especially when considering there is at least one person with a question about filtering methods every time a talk is presented in state and parameter estimation for ice sheets (as the authors have pointed out, there are already others working toward the ice-sheet version of the other main tool of weather prediction, 4Dvar). Thus i feel it is a worthwhile study which should be worthy of publication as a methodological investigation (and the authors frame it as such, at least in the conclusions section. However I do think the manuscript needs work before this can happen. On the basis of the extent of the comments below I choose "major revisions" – but there is no formal definition of what this means, and the editor may choose to ignore this classification. I am not suggesting modification of the algorithm and/or results, simply clarity of text.

I thanks Dan Goldberg for the time spend on trying to understand the description of the method. His comments where very helpful to point parts that where unclear. I totally agree that there is still along road toward operational ice-sheet forecasting, and that will require to *i)* develop/adapt new methodologies but also *(ii)* improve the data bases and the characterisation of the observation uncertainties as pointed in my discussion. Indeed, I see this paper as a methodological investigation to see and present the potential of the method.

Indeed there is several deterministic variants of the Ensemble Kalman Filter. All are based on the Kalman update equation for the analysis. Keeping the covariance matrix in a square root factorisation form, allows to derive an expression for the transformation of the forecast ensemble to the analysed ensemble. As different ensembles could have the same mean and covariance matrix, different implementations will lead to different analysed ensembles and thus different solutions. As mentioned in the text, ESTKF is closely related to the SEIK and ETKF filters that are widely used in oceanography and meteorology. In the original paper that derives the ESTKF, the implementation of ESTKF is estimated to be slightly more efficient that the ETKF, while it shares the same interesting properties of minimal transformation. However the two filters should lead to very similar solutions. I hope that the description of the filter has been clarified in the new version.

**GENERAL COMMENTS**

For one thing upon reading I had significant detail understanding what was done. There were a number of points on which i felt clarity was needed, and most of these are addressed below in line-by-line comments so I will not list them here. Note that these specific questions compose the bulk of my review – and this is because without having a better idea of what was actually implemented, it is difficult to critique the results further!

Thanks. See my point by point answer below.

However something I will state in the general comments is that despite similarities, ice-sheet models are distinct from e.g. atmospheric models in that the unknown parameters most sought cannot generally be observed directly, in contrast to models in which the initial conditions in a forecast/analysis cycle represent the parameters of greatest interest. This is exemplified by the fact that the "state" vector contains non-dynamic variables (friction and bed elevation) and the fact that the observation operator, rather than being a simple averaging or restriction, encapsulates a fully nonlinear solve of an elliptic partial differential equation. I think this is something that should be made very clear to members of the climate and meteorological community who read this work.

It is relatively classical that not all the variables are observed, even for dynamic state variables; and EnKFs are also widely used for state and parameter estimation. I have clarified the fact that the observation operator is the force balance equation and thus highly non-linear..

One overall comment is regarding the distribution of the ensemble. As I understand it, even if the initial ensemble is evenly distributed given the prior, it is difficult to know a pri-ori whether the projection of the ensemble will represent a favorable distribution of the projected space. That is, what if the forecast "clusters", underrepresenting important regions of state space? As I ask below, it is unclear whether there is a "reinitialisation" of the ensemble at every step. Clearly this topic has already been considered in the NWP literature, for example Song et al (2013) makes use of a time-dependent adjoint (a tool the authors state the work here is meant to circumvent the need for) in order to generate a more representative ensemble.

The algorithm leads to a transformation of the forecast ensemble to the new analysed ensemble, that has the mean and covariance matrix given by the original Kalman update equation. Yes with small ensembles, because the covariances matrices are given by a "small" ensemble that can not represent the full error-space, the analysis tends to be over-confident resulting in ensembles with a spread that becomes to small. On the long run the analyses may start to ignore the observations, causing an ensemble that will diverge from the truth. Localisation and inflation are pragmatic remedies to this undersampling issue. Indeed combining ensemble and variational methods will lead to more robust estimation but at an higher cost. I have clarified the presentation of the localisation and inflation methods, and state in the introduction that the tendency is to combine variational and ensemble methods to take the advantages of both. The objective of this first study is to use tools that are widely use in other domains to test their applicability for ice-sheet modelling. Complexity can be added in future studies when the main weaknesses of the method will be clearly identified; so I'm not aiming to give here a detailed discussion on hybrid ensemble/var methods.

The new section 2.2.2 now reads:

**2.2.2 Filter stabilisation: inflation and localisation**

*In practice for large scale problems, EnKFs as Monte-Carlo methods can suffer from under-sampling issues. First, because of the rank deficiency of the covariance matrix $\mathbf{P}^f$, the analysis adjust the model state only in the error subspace, ignoring error directions not accounted for by the ensemble (Hunt et al., 2007). This can result in an analysis that is overconfident and underestimates the true variances. On the long run, the ensemble spread will become too small and the analysis will give to much weight on the forecast finally disregarding the observations and diverging from the true trajectory. A common simple* ad-hoc *remedy is to inflate the forecast covariance matrix with a multiplicative factor (Pham et al., 1998; Anderson and Anderson, 20 199). Here, inflation has been introduced in Eq. (??) using the forgetting factor $\rho \in [0,1]$ with $\rho = 1$ corresponding to no inflation (Pham et al., 1998). It is the inverse of the inflation factor used by Bonan et al. (2014).*
*Second, the rank deficiency of $\mathbf{P}^f$ leads to the appearance of spurious correlations between parts of the system that are far away. As these correlations are usually small, a common remedy is to damp these correlations with a procedure called localisation. In covariance localisation, localisation is applied by using an ensemble covariance matrix that results from the Schur product of $\mathbf{P}^f$ with an* ad-hoc *correlation matrix that drops long range correlations (Hamill et al., 2001; Houtekamer and Mitchell, 2001). However, this localisation technique is not practical for square-root filters where $\mathbf{P}^f$ is never explicitly computed. Here, as in Bonan et al. (2014), we use a localisation algorithm based on domain localisation and observation localisation (Ott et al., 2004; Hunt et al., 2007). Both methods are illustrated in Sakov and Bertino (2011) who conclude that they should yield to similar results. Domain localisation assumes that observations far from a given location have negligible influence. In practice, the state vector in each single mesh node is updated independently during a loop through the nodes that can easily be parallelized for numerical efficiency. For each local analysis, only the observations within a given radius $r$ from the current node are used. In addition to avoid an abrupt cut-off, the observation error covariance matrix $\mathbf{R}$ is modified so that the inverse observation variance decreases to zero with the distance from the node using a fifth-order polynomial function which mimics a Gaussian function but has compact support (Gaspari and Cohn, 1999). Because it drops spurious long-range correlations and allows the local analyses to choose different linear combinations of the ensemble members in different regions, localisation implicitly increases the rank of the covariance matrix, leading to a larger dimension of the error subspace, implicitly increasing the effective ensemble size and the filter stability(Nerger et al., 2006; Hunt et al., 2007). However, it has been reported that localisation could produce imbalanced solutions (Mitchell et al., 2002). Here, because the force balance are non-inertial and the SSA assumes that the ice-shelves are in hydrostatic equilibrium, this shouldn't be an issue. Another disadvantage is that, when long-range correlations truly exist, the analysis will ignore useful informations that could have been used from distant observations.*
*Here, the forgetting factor $\rho$ and the localisation radius $r$ will be used as tuning parameters of the filter. Improving the theoretical understanding of these* ad hoc *procedures and developing adaptive scheme is an active research area and interested readers can refer to review articles ((e.g. Bannister, 2017; Carrassi et al., 2018; Vetra-Carvalho et al., 2018)*

I also question whether the "toy problem" proposed by the authors truly tests all of the difficulties a filtering approach might encounter. I bring this up in more detail below but some aspects of the approach seem to hang on the "locality" of the problem

(a technique called "localisation" is employed to ignore long-distance correlations of the state). I wonder if this only works because the problem is one-dimensional with no buttressing involved, so essentially to a strong degree (though not completely) the velocities depend locally on basal friction and geometry? Would this still be a good approach in a 2D domain with an expansive embayed ice shelf (such as the Ross or FRIS), or very weak basal traction over a large part of the domain (such as Pine Island)?

As explained in the new version, localisation ignores observations that are far away for the local analyses as, often, long range correlation, are poorly estimated due to the sampling. In the case where true long range correlations may exist it will simply ignore this so the analysis may not use the full informations contained in the observations. The lower computational cost due to a small ensemble comes at a price. However I don't think that the 1D experiment ignore this problem. On the contrary, because the shelf is unbuttressed we may expect that the velocities on the shelf are extremely correlated with the basal conditions at the GL. Localisation is now better explained but I agree that more work will be required in the future to see if this can be improved. Following the suggestions from reviewer 2, I know compare the values used in this study with those given by Bonan et al. (2014):

*We have used inflation and localisation to stabilise the filter. The inflation giving the best results in Bonan et al. (2014) ($\rho = 0.87 - 1.02$) is similar to the values tested in this study. For the localisation radius $r$ we have used values between 4 and 16 km, while it ranges from 80 to 120 km in Bonan et al. (2014). While this seems counter-intuitive as the velocities depends only on the local conditions with the shallow ice approximation used by Bonan et al. (2014), in fact, because we use a different grid size ($dx = 0.2km$ compared to $dx = 5km$ in Bonan et al. (2014)), for each node we assimilate twice as much observations. Our results are in agreement with the adaptive localisation radius proposed by Kirchgessner et al. (2014) . Using three different models, Kirchgessner et al. (2014) have shown that good performances are obtained when $r$ is such that the effective local observation dimension, defined as the sum of the weights attributed to each observation during the local assimilation, is equal to the ensemble size. Here, the value $r = 8km$ used for the 50 members-ensemble corresponds to an effective observation dimension of 56. Future studies should investigate if this result can be transposed to realistic 2D simulations with unstructured meshes.*

The methodology essentially uses a whole "family" of geometries and velocities to infer hidden parameters of the system. This somewhat bears similarity to a different paper led by the author, "Assimilation of surface velocities acquired between 1996 and 2010 to constrain the form of the basal friction law under Pine Island Glacier" – aside from the statistical formality, and the introduction of consistency between these geometries by way of the continuity equation (which is actually not so consistent if the analysis updates do not conserve mass!!!) – I wonder if the author would consider comparing and contrasting these approaches.

In the paper you mention, this was a collection of "snapshot" inversions where we assumed a constant sliding coefficient between consecutive observed geometries and indeed we didn't use the continuity equation. It would be interesting to compare both method for the reconstruction part, i.e. the performances for retrieving constant basal properties, but I think this would make a full paper. The advantage of the method used here is that we have, at the end of the assimilation window, a transient model that has been initialised using the available observations and we can make projections. So I prefer to discuss this approach in the context of initialising a transient ice flow model, similarly to 4D-Var, and I think a discussion between these approaches will add more confusion. Concerning the mass conservation, yes the analysis does not conserve mass as the ice-sheet volume is uncertain because the geometry is uncertain. However we expect that the uncertainty is reduced as observations are assimilated and we hope that the volume we have at the end of the assimilation window is a good estimator of the true volume, obtained by the combination of the model and the observations. However, you are right that if we are interested by a re-analysis of the volume change during the assimilation window a 4D-Var or a smoother might be more appropriate. I have tried to clarify this point and hope that the new figure 1 clarifies the concept of sequential DA.

I have added a new paragraph to discuss this point in section 3.3 Assimilation set-up:

*Because both $z_s$ and $b$ are included in the state vector, the analysis does not conserve the ice sheet volume, neither for the ensemble mean and the individual members. However, the estimation of the ice-sheet volume is improved at each analysis as more data are assimilated, and the final volume is the best estimation provided by the filter knowing the model, all the observations during the assimilation window and their uncertainties. As mentioned in the introduction, if the main interest is an analysis of past volume changes, as smoother might be more appropriate and the smoother extension of the ESTKF can be found in Nerger et al. (2014).*

The distinction between EnKF, 4DVar and smoother is introduced in the introduction as:

*EnKF approximates the state and the error covariance matrix of a system using an ensemble that is propagated forward in time with the model, avoiding the computation of the covariance matrices and the use of linearised or adjoint models. Contrary 15 to time-dependent variational methods where the objective is to find the model trajectory that minimizes the difference with*
5 *all the observations within an assimilation window, EnKF assimilates the observations sequentially in time as they become available using the analysis step of the Kalman Filter, as illustrated in Fig. 1. The model trajectory is then discontinuous and, at a given analysis, the model is only informed by past and present observations. For the retrospective analysis of a time period in the past, i.e. a reanalysis, ensemble filters can easily be extended to smoothers to provide analyses that are informed by all 20 past, present and future observations (Evensen and van Leeuwen , 2000; Li and Navon, 2001; Cosme et al., 2012; Nerger*
10 *et al., 2014)*

Finally, i point out that, despite the divide between filtering and adjoint-based methods, there is a growing sentiment in NWP to take what is "best" from the various approaches and form more hybrid schemes (for instance the Song paper referenced above, see also Kalnay 2010). Therefore I urge the author to reflect on such innovations and how they might be useful in further developments for filtering of ice-sheet models.

15 I now mention this in the introduction, but I think first we have to test the performances of individual methods to identify the weak points, so going to far in a discussion of what we might expect by using hybrid methods is premature.

**Line by Line comments**

– p2.l3: would be good to state this is a point when only resolving 1 horizontal dimension

this has been changed to:

20 *Improving SLR estimates requires, amongst others, to correctly model the dynamics of the grounding line (GL), i.e. the location where the ice detaches from its underlying bed and goes afloat on the ocean, implying that this is a line in 2D-plane view and a point in 1D..*

– p3.10 variational

done

25 – p3.13 "use of linearised or adjoint models" this assumes a trivial mapping from model vars to observations – see my comments below.

This point has been clarified. If it is true that the linearised observation operator is used in the KF update equation, in practice we do not need to compute it as it always act as an operator to project the ensemble members in the observation space, and we make the classical linear approximation

30 $$\mathbf{Y}^f = \mathbf{H}\mathbf{X}^f \qquad (1)$$

with $\mathbf{X}^f = (\boldsymbol{x}_1^f, ..., \boldsymbol{x}_{N_e}^f) \in \mathbb{R}^{N_x \times N_e}$ the forecast ensemble matrix and $\mathbf{Y}^f = (\boldsymbol{y}_1^f, ..., \boldsymbol{y}_{N_e}^f) \in \mathbb{R}^{N_y \times N_e}$ its equivalent in the observation space where $\boldsymbol{y}_i^f = \mathcal{H}(\boldsymbol{x}_1^f)$, $i = 1, ..., N_e$. This point has been clarified in section 2.2

– p3.15 rewrited?

yes thanks.

35 – p3.35 – would be good to explain as soon as possible i.e. here what you mean by a twin experiment, or give a reference, as this is jargony

The meaning of twin experiment is now introduced in the introduction:

*In the context of ice-sheet modelling, encouraging results have been obtained by ? for the estimation of the state and basal conditions of an ice-sheet model using the Ensemble Transform Kalman Filter (ETKF, ??). They study the performance*
40 *of the method using idealised twin experiments where perturbed observations generated from a model run are used in the DA framework to retrieve the true model states and parameters.*

- p 5 thru eq (10): this is a well written explanation of the EnKF. However I have a few questions which might be due to my lack of familiarity with filtering methods, but I think this might be true of many readers of this paper. This is also important as, though it is not the algorithm used, the one used is far more complex so this is a chance to explain your methods to the reader.

  (a) is $P\hat{}f = P_k$?

  *Yes I mention that I omit the time index $k$ as all the analysis is done at a given time $t_k$. However, ESTK rewrites $P^f$ as a function of $X^f$ and $\Omega$ as explained in the new version.*

  (b) you do not say how the individual ensemble members ($x_i\hat{}k, a$) arise/are updated, only the state vector (which looks like the mean of the analysed ensemble)?

  *it is now clear that the update equation for the mean and covariance is rewritten to give a unique equation for the transformation of the forecast members $x_i^f$ to the analysed members $x_i^a$*

  (c) is the posterior/analysis covariance used at all in subsequent time/filtering steps, as from eq 7 the covariance is always formed from the present ensemble – so i am struggling to grasp what is done in the algorithm in a multi-time step (k>1) framework.

  *I now clearly mention that the analysed ensemble is used as the initial ensemble for the next forecast, and so on until the end of the assimilation window. I also think that the new figure 1 helps to clarify this point.*

  (d) For each new forecast/analysis cycle, is the ensemble generated anew from the analysis-generate ensemble statistics?

  *See reply above, the analysed ensemble is directly obtained for the analysis equation and can be used as the initial ensemble for the next forecast.*

  (e) the formula given assumes normality of the ensemble does it not?

  *As mentioned in the introduction the KF analysis is optimal for Gaussian distributions and it only uses the first moments of the distribution, mean and covariance.*

- P5 eq(8): $M_k$ is trivially the identity on the time-invariant components of $x$, i.e. $b$ and $C$, correct?

  *Yes, I now explicitly mention that I assume persistence for the parameters during the forecast step, in sec 3.3:*
  *The state vector is augmented by the two parameters to be estimated, the bedrock topography b and the basal friction coefficient C. For the parameters we assume a persistence model, i.e. no time evolution, during the forecast step (Eq. 8).*

- p6.4. I am struggling to see why $P\hat{}f$ need be formed, as it is a tensor product of $X$ with itself (subject to (a) above). For instance, the last term in 9(a) is written

$$X(X\hat{}TH\hat{}T)((HX)(X\hat{}TH\hat{}T) + R)\hat{} - 1 \tag{2}$$

so the largest matrix that need be formed is HX, and no matrix of (Nx x Nx) need be formed. Perhaps I do not understand where and how P^a is actually used however.

*Yes this is the basis of the implementation, because the covariance matrices can not be formed, we keep the square-root factorisation. This allows to obtained an expression for the transformation of the forecast ensemble to the analysed ensemble that exactly has the covariance matrix $P^a$. So we don't have to explicitly compute and store $P^a$.*

- P6.5 I don't feel the concept of "error subspace" is ever suitably explained as i read the paper still wondering about this. $\Omega$ as defined in eq 11 simply seems to be a "mixing" matrix that slightly changes the ensemble members – how is this an "error subspace"? (Assumiing that $X \in R\hat{}NxxNe$ – i take it this is the case for eq 11 to make sense...)

*This is now explained as follow:*
*Moreover, the sample covariance matrix approximated with an ensemble of size $N_e$ (Eq. 7) is only a low-rank approximation of the true covariance matrix and its rank is at most $N_e - 1$. ESTKF uses this property to write the analysis in*

*a $(Ne - 1)$-dimensional subspace spanned by the ensemble and referred to as the error subspace (Nerger et al., 2005a) The forecast covariance matrix $\mathbf{P}^f$ is then rewritten as*

$$\mathbf{P}^f = \frac{1}{N_e - 1} \mathbf{L}\mathbf{L}^T \tag{3}$$

*where $\mathbf{L} \in \mathbb{R}^{N_x \times N_e - 1}$ is given by*

$$\mathbf{L} = \mathbf{X}^f \mathbf{\Omega} \tag{4}$$

*The matrix $\mathbf{\Omega} \in \mathbb{R}^{N_e \times N_e - 1}$ defined as*

$$\Omega_{ij} = \begin{cases} 1 - \dfrac{1}{N_e} \dfrac{1}{\frac{1}{\sqrt{N_e}} + 1} & \text{for } i = j, \ i < N_e \\[2mm] -\dfrac{1}{N_e} \dfrac{1}{\frac{1}{\sqrt{N_e}} + 1} & \text{for } i \neq j, \ i < N_e \\[2mm] -\dfrac{1}{\sqrt{N_e}} & \text{for } i = N_e \end{cases} \tag{5}$$

*projects the ensemble matrix $\mathbf{X}^f$ onto the error subspace. The multiplication with $\mathbf{X}^f$ subtracts the ensemble mean and a fraction of the last column of the ensemble perturbation matrix $\mathbf{X}'^f$ from all other columns.*

– Eqs 11-16: in contrast to the discussion of the EnKF this is very nonintuitive. You state (P6 line 7) that you approximate the covariance matrix by a low-rank matrix, which seems intuitive, but where is the equation describing this low-rank approximation and how is it done? (for what it is worth, low-rank approximations of covariance generally involve eigenvalue decompositions to retain the leading order covariance structure, but i do not see this here...

See reply above. It is a low rank matrix as it is approximated by an ensemble of size $N_e$, so the rank of the sample covariance matrix is at most $N_e - 1$, while, in principle, the true covariance matrix could be full-rank.

– P6.23-25: can you give a more intuitive description of inflation? Why do you need it and what does it achieve? As it is I am not even sure if inflation corresponds to lower or higher $\rho$.

Following suggestions from reviewer 2, I now give better explanations of inflation and localisation in section 2.2.2. See reply above.

– P7, first paragraph. I'm sorry but I am struggling to follow this paragraph. For instance, how does the non-linear observation operator applied to $x_i$ lead to the product $HX^f$? i imagine they are related, as $H$ is a linearisation of $\mathcal{H}$ (which, by the way, clashes with the symbol for ice thickness) but this is not explained.

See reply above. This is now better explained and I use $\mathcal{H}$ for the non-linear observation operator.

– P7.8. This assumption, i imagine, is valid in many NWP settings given the hyperbolicity of the equations. Are you confident it is a good approach for marine ice-sheet modelling?

Also in NWP you may have long-range correlations. Localisation is better explained and presented as a pragmatic way to counteract under-sampling issues. As you mention this paper is a first step toward operational DA and more work will be required on these aspects.

– P8.26. I was surprised by your suggestion that annual DEMs would be available over a multiple decadal period, as i do not know of such products for antarctica. The best i have seen is decadal or semidecadal with MUCH lower spatial resolution (e.g. Konrad et al 2017, GRL). Having skimmed the ArcticDEM website i do see mention of the spatial resolution, but not the temporal. Unless you can argue that such spatiotemporal resolution is reasonable and available, i suggest caveating this discussion by saying it is an idealised experiment and this is the type of spatiotemporal resolution to which the community should aspire.

I agree that we don't have this kind of product yet for decadal periods, but it should become available quickly and you can now find DEMs for the Greenland Ice sheet with a with a 3-month temporal resolution (https://nsidc.org/data/nsidc-0715). This has been updated.

– P9.6: prognostic ice sheet models generally step forward the ice thickness, not surface elevation (as shown in your eq 4). In the analysis step you are updating $z_s$. Is there a simple mapping from your model state X to thickness?

Yes I now clearly mention that the floatation equation is used for the mapping between the ice thickness and the free surface elevation. As mentioned in the discussion, with a full-stokes model that solves the two free surfaces you will have to put both free surfaces as state variables.

– P9.6: as mentioned in the previous comment you are updating $z_s$ in each analysis step, which i am inferring then maps on to an update in thickness (tell me if i am wrong). Is this update at all volume conserving? If not should this be a concern?

See reply above. Because this is a sequential algorithm the analysis does not conserve volume because the volume is uncertain, i.e. both the free surface and bed elevation are in the state vector. This is not a concern as this is the aim of DA to improve the estimation of the system state by using the observations. Again, for a re-analysis it might be more appropriate to use a smoother instead of a filter to interpret past volume changes.

– P9.21-28: Lots of jargony language in this paragraph, likely not to be understood by the target audience. What is a sill and a nugget? You talk of the prediction obtained by kriging – is this something you have calculated? Is there any way to evaluate whether the ensemble does converge to it? Is this a way of evaluating whether the ensemble is large enough?

It is better explained now. The method directly draw realisations, i.e. members, from the distribution that would be obtained from kriging. This is not an exact sampling method, i.e. the mean and covariance of the ensemble will not exactly match the mean and covariance of the kriging prediction. As the spatial correlation is already given by a model, i.e. the analytical variograms, we want an initial ensemble that is representative of what we think is the initial uncertainty given the available a-priori or observations, and this is what we get.

– Section 4.1: Upon reading this, I realised that (a) i am unsure what time step you used, and (b) more importantly, whether each time step is a forecast/analysis step, as $M_k$ in eq 8 could easily encompass multiple time steps – is this the case?

I now mention that the time step is 0.005 a and the time interval between two analysis is 1 year. So yes it is clear now that we can have multiple model time steps between two analyses.

– P10.15: again, these factors seem very important, and as mentioned above not overly well explained.

see reply above

– Section 4.2 – section headings should be capitalised

Done

– P12.25: Two comments about this paragraph: (a) Code that is continuously being up- dated and new algorithms developed might be an issue for *analytically* derived adjoint models, but not as much for automatic differentiation, which is specifically designed to generate new adjoint code when the "primal" code is changed; and (b) I return to the my confusion over the first paragraph on P7. Your observation matrix $H$ contains, at the very least, a linearisation of the stress balance equation mapping geometry and basal friction onto velocity. It is not clear how you are finding this operator if not through some sort of forward model linearisation.

ok I have removed this sentence. I agree that with automatic code differentiation, in principle it should be relatively straightforward; however, I think most people will still agree that maintaining an adjoint code remains difficult in practice for research codes. As explained above there is no linearisation required in the method used here.

**0.1  References:**

Song et al, 2013: An adjointbased adaptive ensemble Kalman filter. Mon. Wea. Rev., 141, 3343–3359, https://doi.org/10.1175/MWR-D-12-00244.1.

Kalnay, E. (2010), Ensemble Kalman Filter: Current status and potential, in Data Assimilation: Making Sense of Observations,
5   edited by W. Lahoz, B. Khattatov, and R. Menard, 24 pp., Springer, New York

**ANONYMOUS REVIEWER 2**

First of all, I would like to apologise to the author and the editor for providing my review so late. I consider this paper treats an important subject in an innovative manner and, as such, deserves a careful review. I hope you will find my review insightful. Yes many thanks for this insightful review. Please find below my point by point answer.

10   **OVERVIEW**

The paper aims to adapt an Ensemble Kalman Filter (EnKF) to estimate jointly the surface elevation, the bedrock topography and the basal friction coefficient in the case of a flowline marine ice sheet. Time-dependant ensemble data assimilation (DA) approaches are relatively new for ice sheet initialisation. The paper focuses on the case of a grounding line retreat for unstable glaciers, which is a hot topic in ice sheet modelling and climate change. It also studies the influence of DA on forecasts of
15   grounding line retreat.

**GENERAL COMMENTS**

The paper targets an important and timely question: how to initialise and estimate basal parameters to forecast the evolution of marine ice sheets and glaciers especially in the case of an unstable retreat? and proposes to use an EnKF (here an ESTKF) in this context. EnKFs have shown how efficient they can be in a wide range of applications (not exclusively meteorology and
20   oceanography but also hydrology, crop modelling, oil extraction, pollutant dispersion, . . .), are a good alternative to adjoint-based methods and are the basis of hybrid methods that are now popular in DA (see e.g. Bannister, 2017). The paper shows clearly that EnKFs are a good toolbox for DA in glaciology. The experiment shows clearly how beneficial this approach could be with adequate figures and a very insightful analysis. Overall I am convinced that the paper in its final form will be a very good introduction of ensemble DA methods in ice sheet modelling.
25   Nevertheless, there are few points that prevent me to publish the paper as is. I list them below:

- I have some reservation about the experimental setup mainly on how the reference run is designed and on how the basal friction coefficient $C$ is estimated.

  - About the reference run: The long-term objective of the study is to forecast accurately grounding line retreat for Antarctic hemispheric glaciers in the context of global change. However, in the reference run of the experiment,
30     the grounding line retreat is triggered by the abrupt change of ice rigidity $B$. I wonder if the simulated grounding line retreat is "realistic" compared to one triggered by climate change. Also I have the same question about the sinusoidal basal friction parameter $C$. How realistic is it compared to real cases? I know we are in flowline cases using SSA equations. But it would strengthen the paper if the author could reflect on that subject in the section 3.1. Part of my comment may be due to my lack of knowledge in glaciology, so please accept my apologies if my
35     comment is irrelevant.

    I now compare the synthetic values for $b$ and $C$ with values obtained by Brondex et al. (2018) in Thawaites Glacier Antarctica (Fig. 2). This shows that the values are realistic. Note however that the mesh resolution in Brondex et al. (2018) varies from ≈ 200m close to the GL to 10km 100km upstream, so that the comparison for the wavelengths are really meaningful only in the first tens of kilometres. For the initial retreat I have added the following discussion:
40     *In Jenkins et al. (2018), observed ice-flow accelerations in the Amundsen sea sector have been attributed to the*

*decadal oceanic variability, where warm phases associated with increased basal melt induce a thinning of the ice shelves reducing their buttressing effect initiating short lived periods of unstable retreat of the most vulnerable GLs. In a flow line experiment the ice shelf do not exert any buttressing effect. Using a suite of melting and calving perturbation experiments for Pine Island Glacier, Favier et al. (2014) have shown that, when initiated, the dynamics of the unstable retreat is fairly independent of the type and magnitude of the perturbation. Here, to trigger the initial acceleration, we instantaneously decrease the ice rigidity to $B = 0.3\,MPa\,a^{-3}$ at $t = 0$, keeping all the other parameters constant.*

[Figure]

**Figure 2.** NEW FIGURE 4. Thwaites Glacier (Antarctica). Model results from **?**: model velocities (top) and friction coefficient $C$ and bed elevation $b$ extracted along three streamlines (same color code). Synthetic values used in this study are shown with black dashed lines. Note that the mesh resolution varies from ⌣ 200m close to the GL, shown in yellow in the top panel, to ⌣ 10km at the upstream end of the streamlines.

    – About the assimilation setup: The basal friction coefficient $C$ must be positive. To ensure such thing, you use the following change of variable $C = \alpha^2$. But by doing so, you do not ensure the unicity of the estimation as $\alpha$ and
10     $-\alpha$ would lead to the same $C$. Also I thought that the $C$ parameter could be of different order of magnitude. To counteract those potential issues, Bonan et al. (2014) chose the following change of variable $C = 10^\alpha$ ensuring the positivity of $C$, the unicity of the change of variables and mimicking behaviours with different order of magnitude. Could you explain why the change of variable you used is more appropriate in your context?

    *There was no particular reason, only that the reference only span one order of magnitude. To my knowledge I*
15     *was one of the first to introduce the change of variable $C = 10^\alpha$. I have performed a new experiment with this change of variable to see the differences. The results a given in Figure 3. It can be seen that the performances are extremely similar. However, for $C$ the RMSE is a little higher (0.0045, instead of $0.004\,Mpa\,m^{-1/3}\,a^{1/3}$), while there is nearly no difference for the velocities. I think this can be explained by the fact that the sensitivity of the model velocities to the remaining uncertainty in $C$ is lower that the observation uncertainty so the DA can not*
20     *really discriminate the two reconstructions. To see this effect I present in the new version of the paper, additional*

experiments where I study the error on the retrieved values as a function of the observation noise. see answers below.

The part describing the change of variable as been updated as follow:

*Because the velocities are insensitive to the basal conditions where ice is floating, these two parameters are included in the state vector only for the nodes where at least one member is grounded. In addition, to insure that $C$ remains positive, we use the following change of variable for the assimilation $C = \alpha^2$. Although it does not insure uniqueness of the estimation as $\alpha$ and $-\alpha$ would lead to the same $C$, this change of variable is classical (Mac Ayeal (993) and was chosen as the reference friction coefficient spans only one order of magnitude. Similar performances where found using the other classical change of variable $C = 10^\alpha$ as in Gillet-Chaulet et al. (2014).*

[Figure]

**Figure 3.** RMSEs after analysis for the assimilation up to $t = 35a$. Results shown in the paper with he change of variable $C = \alpha^2$ are in red; results with the change of variable $C = 10^\alpha$ are in blue.

- The paper is full of interesting results but sometimes they deserve a more thorough analysis.

  – The assimilation window is between 1 yr and 35 yr (you run forecasts from analysed states at t = 20 yr and t = 35 yr). But the grounding line is almost steady between t = 13 yr and t = 32 yr. We also see that after the first 10 years of assimilation, RMSD for $C$ remains stable. I wonder if those two points are correlated meaning is it easier or more difficult to estimate $C$ in the case of a retreating grounding line (hence a more dynamic ice sheet) or an almost steady state? As the behaviour of grounding line is different from one marine glacier to another, it would be beneficial to the community if you could push your study in that direction in the revised version of the manuscript (for example continuing DA after t = 35 yr for the next ten years and study what happens).

    We don't see more improvement if the assimilation is pursued up to $t = 50a$. My understanding is that the sensitivity of the observations to the remaining uncertainties is already below the noise level, especially for the velocities, so we still see an improvement for the few kilometres upstream of the grounding line but this do not reflects on the values for the RMSE for $C$ and $b$, that are computed from $x = 300km$ up to the position where at least one member is grounded (Fig. 4 and 5). However, we still see an improvement for the forecast as shown in Figs. 6 and 7

[Figure]

**Figure 4.** RMSEs after analyses for an assimilation up to $t = 50$a.

[Figure]

**Figure 5.** Same as Fig. 3 but or an assimilation up to $t = 50$a.

[Figure]

**Figure 6.** Same as Fig.2 but for an assimilation up to $t = 50$a.

[Figure]

**Figure 7.** Same as Fig. 12 but or an assimilation up to $t = 50$a.

I have added two additional experiments where I vary the noise level for the observed velocities and the surface elevation (Figures 8 and 9). This shows that decreasing the uncertainty on the observed velocities improves the RMSE for $b$ and $C$, but the results for the reconstructed velocities are not significant, so that 2 ensembles with slightly different RMSE have very similar differences for the velocities and the reconstruction stagnates. Changing the noise for the observed surface elevation has a small effect and in fact $RMSE_C$ increases for the lowest noise levels but this do not reflect on the velocities.

[Figure]

**Figure 8.** NEW FIGURE 10.Sensitivity to the surface velocities observation error $\sigma_u^{obs}$ : RMSEs after each analysis, computed only for $x \geq 300km$ for $b$ and $C$.. The thick red lines correspond to the results with $\sigma_u^{obs} = 20$ ma$^{-1}$ and $\sigma_{z_s^{obs}} = 10$ m shown in Figs. **??** and **??**. The horizontal dashed lines correspond to the observation errors $\sigma_u^{obs}$ and the results are presented with solid lines using the same color code. $\sigma_{z_s^{obs}} = 10$ m for all the experiments.

[Figure]

**Figure 9.** NEW FIGURE 11. Sensitivity to the surface elevation observation error $\sigma_{z_s^{obs}}$: RMSEs after each analysis, computed only for $x \geq 300km$ for $b$ and $C$. The thick red lines correspond to the results with $\sigma_u^{obs} = 20$ ma$^{-1}$ and $\sigma_{z_s^{obs}} = 10$ m shown in Figs. **??** and **??**. The horizontal dashed lines correspond to the observation errors $\sigma_{z_s^{obs}}$ and the results are presented with solid lines using the same color code. $\sigma_u^{obs} = 20$ ma$^{-1}$ for all the experiments.

– Figure 5 shows interesting results about the performance of ESTKF with inflation and localisation and varying sizes of the ensemble. Bonan et al. (2014) has performed the same kind of studies for grounded ice sheet using an ETKF. Do you obtain similar optimal parameters for inflation and localisation in your experiment or are they different from Bonan et al. (2014)? It would be interesting to reflect on how the physics of the model influences such parameters.

*I have added the following discussion:*

*We have used inflation and localisation to stabilise the filter. The inflation giving the best results in Bonan et al. (2014) ($\rho = 0.87 - 1.02$) is similar to the values tested in this study. For the localisation radius $r$ we have used values between 4 and 16 km, while it ranges from 80 to 120 km in Bonan et al. (2014). While this seems counter-intuitive as the velocities depends only on the local conditions with the shallow ice approximation used by Bonan et al. (2014), in fact, because we use a different grid size ($dx$ = 0.2km compared to $dx$ = 5km in Bonan et al. (2014)), for each node we assimilate twice as much observations. Our results are in agreement with the adaptive localisation radius proposed by Kirchgessner et al. (2014). Using three different models, Kirchgessner et al. (2014) have shown that good performances are obtained when $r$ is such that the effective local observation dimension, defined as the sum of the weights attributed to each observation during the local assimilation, is equal to the ensemble size. Here, the value $r = 8km$ used for the 50 members-ensemble corresponds to an effective observation dimension of 56. Future studies should investigate if this result can be transposed to realistic 2D simulations with unstructured meshes.*

– You only show results after 300 km (nothing between 0 km and 300 km). Does it mean that what happens between 0 km and 300 km does not have an influence on the grounding line retreat? Does it mean DA is pointless in those areas (in that case, that would make DA more affordable as less grid points need to be treated)? If so, please state it more clearly and if not, provide more results for that area.

Yes as shown by Durand et al. (2011), we expect that uncertainties in the ice-sheet interior should not affect short-term forecast of the coastal regions. However for completeness, I have added a figure that shows the results in the

first 300km. Because the noise level on the observed velocity is close to 100% the is only little improvement for $b$. It is better for $C$. Also the ensemble spread is only slightly reduced. My understanding is that the sensitivity of the observations to the initial uncertainty is smaller than the noise level.

[Figure]

**Figure 10.** NEW FIGURE 8. Same as Fig.3. but with the RMSE computed for $x \in [0, 300]$ km.

– I am worried by some of the results shown for ESTKF for small ensembles (Ne = 30) as RMSD reduction is only 20% for the bedrock topography and 30% for the basal friction coefficient. For 2D real cases (either using Full-Stokes or SSA), we need an ensemble approach working for small ensembles due to the cost of the experiment and 30 members might be too expensive. Could you reassure me and provide the reader that the approach would be beneficial even in 2D real cases?

Results are similar to Bonan et al. (2014), the performances start to deteriorate with ensemble sizes $\leq 50$. However, I'm optimistic that we shoul be able to run 2D application, at least at the scale of a drainage bassin with ensemble sizes $N_e$ at the order of 50 to 100. See the following sentences in the discussion:

*Good results have been obtained with relatively small ensembles (50 to 100 members) for a state vector of size $N_x \approx 8400$ and $N_y = 4002$ observations. Similarly to Bonan et al. (2014), we still see an improvement with a 30-members ensemble but the performances to retrieve the basal conditions are not as good. Running 2D plane view simulations with such ensemble sizes is largely possible as demonstrated by Ritz et al. (2015) who, using an hybrid shallow ice-shallow shelf model, have run a 200 years ensemble forecast of the whole Antarctic Ice Sheet using 3000 members.*

– About the forecast experiments, Figure 1 shows clearly how beneficial an ensemble forecast can be compared to a deterministic forecast. It also shows that the distribution of the grounding line position is not Gaussian. Could you provide more information (maybe using histograms) on how grounding lines are distributed in the ensemble?

Thanks for the suggestion; I now give the following figure that shows the histograms from the relative volume above floatation (VAF; i.e. what matters when looking at the contribution of ice sheets to sea level rise) and GL position.

[Figure]

**Figure 11.** Ensemble forecast at $t = 100$ a: (top) relative change of volume above floatation (VAF) and (bottom) GL position with (left) no assimilation, (center) assimilation up to $t = 20$ a and (right) assimilation up to $t = 35$ a. The red circle correspond to the reference run and the magenta square to the deterministic forecast.

- While on average, the paper is perfectly readable. There are few parts that are hardly accessible to the reader. As I think the target audience of this paper is the ice sheet community and more generally the glaciology community, the paper would benefit from a careful editing. I detail those parts in the Specific comments section.

  Thanks for these specific comments; see the specific replies below.

5    Overall I consider the paper as a highly valuable addition to data assimilation for ice sheets. But it deserves mostly some rewriting. Therefore I recommend a minor revision as almost all the science is already here!

**SPECIFIC COMMENTS**

Overall I think it would be nice for the reader to have two tables summarising the various variables used in this paper, one for the ice flow model and one for data assimilation.

10    Good suggestion; I now have the following two tables in appendix

**Table 1.** NEW APPENDIX TABLE A1: Notations and values used in this study associated with the ice flow model

| | | |
|---|---|---|
| Prognostic variables: | | |
| $H = z_s - z_b$ | m | Thickness |
| $z_s$ | m | top surface elevation |
| $z_s$ | m | bottom surface elevation |
| Diagnostic variable: | | |
| $u$ | $\mathrm{m\,a}^{-1}$ | horizontal velocity |
| Parameters: | | |
| $a_b = 0.0$ | $\mathrm{m\,a}^{-1}$ | basal melting |
| $a_s = 0.5$ | $\mathrm{m\,a}^{-1}$ | surface accumulation |
| $b$ | m | bed elevation |
| $B = 0.4$ | $\mathrm{Mpa\,a}^{1/3}$ | ice rigidity |
| $C$ | m | basal friction coefficient |
| $m = 1/3$ | | friction law exponent |
| $n = 3$ | | Glen's creep exponent |
| $\rho_i = 900$ | $\mathrm{kg\,m}^3$ | ice density |
| $\rho_w = 1000$ | $\mathrm{kg\,m}^3$ | sea water density |
| Numerical parameters: | | |
| $dt = 5\,10^{-3}$ | a | model time step |
| $dx = 200$ | m | mesh resolution |

**Table 2.** NEW APPENDIX TABLE A2: Notations and values used in this study associated with the ensemble filter

| | | |
|---|---|---|
| Variables: | | |
| $\boldsymbol{x} = (z_s, b, C)$ | | state vector |
| $\mathbf{P}$ | | covariance matrix |
| Stabilisation parameters: | | |
| $r$ | m | localisation radius |
| $\rho$ | | forgetting factor |
| Sizes: | | |
| $N_e$ | | ensemble size |
| $N_x$ | | state vector size |
| $N_y$ | | observation vector size |
| Others: | | |
| $\Delta t = 1$ | a | time interval between two analyses |

About Ensemble Kalman Filters:

- I feel localisation and inflation should be better explained in the paper (either in the introduction and in the DA section). I agree they are both used to counteracts the effects of undersampling. But the reader would benefit from having more information. Undersampling causes underestimated variances (counteracted by inflation) and spurious correlations (counteracted by localisation in the case of long-range spatial spurious correlations). Could you add few lines on the subject. Also few references are missing. For inflation:

  Anderson, J. L. and Anderson, S. L.: A Monte Carlo implementation of the nonlinear filtering problem to produce ensemble assimilations and forecasts, Mon. Weather Rev., 127, 2741–2758, doi: 10.1175/1520- 0493(1999)127<2741/AMCIOT>2.0.CO;, 1999.

  For localisation, the first one is for local analysis (the one you use), the other two are for covariance localisation (the historical one):

  Ott, E., Hunt, B. R., Szunyogh, I., Zimin, A. V., Kostelich, E. J., Corazza, M., Kalnay, E., Patil, D. J., and Yorke, A.: A local ensemble Kalman filter for atmospheric data assimilation, Tellus A, 56, 415–428, doi: 10.1111/j.1600-0870.2004.00076.x, 2004.

  Hamill, T. M., Whitaker, J. S., and Snyder, C.: Distance-dependent fil- tering of background error covariance estimates in an ensemble Kalman filter, Mon. Weather Rev., 129, 2776–2790, doi: 10.1175/1520- 0493(2001)129<2776/DDFOBE>2.0.CO;, 2001.

  Houtekamer, P. L. and Mitchell, H. L.: A sequential ensemble Kalman filter for atmospheric data assimilation, Mon. Weather Rev., 129, 123–137, doi: 10.1175/1520-0493(2001)129<0123/ASEKFF>2.0.CO;, 2001.

  *I now have a section 2.2.2 Filter stabilisation: inflation and localisation where inflation and localisation are better described using the given references. See reply to reviewer 1.*

- Also about inflation, the term "forgetting factor" introduced by Pham et al. (1996) is, unfortunately, very uncommon in the EnKF community. Could you state somewhere that this is just the inverse of the traditional inflation parameter known widely in the EnKF community?

  *I now state that the forgetting factor $\rho$ is the inverse of the inflation factor used by Bonan et al. (2014).*

- p. 6, l. 23: There is no unicity of the symmetric square root matrix C. It is known that the choice of C can have a significant impact on results (see e.g. Livings et al., 2008). Could you detail how C is calculated in PDAF?

  Livings, D. M., Dance, S. L., and Nichols, N. K.: Unbiased ensemble square root filters, Physica D, 237, 1021–1028, doi: 10.1016/j.physd.2008.01.005, 2008.

  *I now mention that $\mathbf{C}$ is the symmetric square root of $\mathbf{A}$ obtained by singular value decomposition. I have added a remark to say that $\mathbf{C}$ could also be computed from a Cholesky decomposition.*

- p. 7, first paragraph on how to use the ESTKF with a nonlinear observation operator. It is a very good point you raise especially in the case of assimilating surface ice velocities (highly nonlinear observation operator). However, I find it difficult to see where the nonlinearity of the observation operator intervenes. Could you rewrite the whole section 2.2 and consider directly the case when the observation operator is nonlinear? That would avoid confusion for readers.

  *Section 2.2 has been updated. I now use $\mathbf{Y}^f = (\boldsymbol{y}_1^f, ..., \boldsymbol{y}_{N_e}^f) \in \mathbb{R}^{N_y \times N_e}$ for the ensemble projected in the observation space with $\boldsymbol{y}_i^f = \mathcal{H}(\boldsymbol{x}_1^f)$, $i = 1, ..., N_e$. The formulas are now given directly using $\mathbf{Y}^f$, however for clarity I still use the linearised observation operator $\mathbf{H}$ for the formula of the Kalman filter update:*

$$\mathbf{K} = \mathbf{P}^f \mathbf{H}^T (\mathbf{H} \mathbf{P}^f \mathbf{H}^T + \mathbf{R})^{-1} \tag{6}$$

*Here, $\mathbf{H}$ is the linearised observation operator at the forecast mean. However, in practice $\mathbf{H}$ does not need do be computed as it always acts as an operator to project the ensemble members in the observation space. Defining the*

*forecast ensemble projected in the observation space by $\boldsymbol{y}_i^f = \mathcal{H}(\boldsymbol{x}_1^f)$, $i = 1, ..., N_e$ with $\bar{\boldsymbol{y}}^f$ the ensemble mean, we make the linear approximation*

$$\mathbf{Y}^f = \mathbf{H}\mathbf{X}^f \tag{7}$$

*with $\mathbf{X}^f = (\boldsymbol{x}_1^f, ..., \boldsymbol{x}_{N_e}^f) \in \mathbb{R}^{N_x \times N_e}$ the forecast ensemble matrix and $\mathbf{Y}^f = (\boldsymbol{y}_1^f, ..., \boldsymbol{y}_{N_e}^f) \in \mathbb{R}^{N_y \times N_e}$ its equivalent in the observation space.*

About the experiment:

– p. 8, l. 1-2: About the roughness signal br ,could you detail, in annex for example, how do you simulate the roughness (which equation?) because I do not know this approach.

I have added a reference to Fournier et al. (1982) where the original algorithm can be found and added the following information in the main text:
*This is a classical algorithm for artificial landscape generation. In 1D, the algorithm recursively subdivide a segment and a random value drawn from a normal distribution $\mathcal{N}(0, \sigma^2)$ is added to the elevation of the midpoint. The standard deviation $\sigma$ is decreased by a factor $2h$ between two recursions. Here we have used 12 recursions using an initial standard deviation $\sigma = 500$ m and a roughness $h = 0.7$.*

– p. 9, l. 21-22: you mention the term "variogram" which is not well known for readers. Could you provide more details how do you generate the ensemble of initial bedrock topographies, in annex for example?

– p. 9, l. 30-31: same comment as previous.

For the two points above, I now introduce the definition of a variogram and give the formulas for the variograms used in this study:
*Following previous studies (Gudmundsson and Raymond, 2008; Pralong and Gudmundsson, 2011; Bonan et al., 2014; Brinkerhoff et al., 2016), we assume that the initial distributions for $b$ and $C$ are Gaussian with a given mean and a prescribed covariance model. Furthermore we assume no cross-correlation between the initial $b$, $C$ and $z_s$ and we draw the initial ensembles independently. For $b$ and $C$, the initial samples are drawn using the R package gstat (Pebesma and Wesseling, 1998). As classical in geostatistics, the covariance model is prescribed using a variogram $\gamma(d)$ that is half the variance of the difference between field values as a function of their separation $d$. It is usually defined by two parameters, the sill $s$ that defines the semi-variance at large distances and the range $r_a$ which, for asymptotic functions, is defined as the distance where the $\gamma(r_a) = 0.95s$. The package gstat allows directly to draw simulations, i.e. random realisations of the field, from the prescribed spatial moments (Pebesma and Wesseling, 1998).*

About the discussion:

– p. 12, l. 25-29: you seem to oppose ensemble and variational methods, but more and more, the tendency is to develop hybrid methods as detailed in Bannister (2017) and Vetra-Carvalho et al. (2018). The main tendency is to use variational approaches in which the adjoint is replaced by ensembles making those adjoint- free approaches. Could you modify your paragraph to reflect this tendency?

I have removed this part from the conclusion, but added a sentence in the introduction:
*Ensemble DA methods, based on the ensemble Kalman filter (EnKF), have been successful in solving DA problems with 10 large and non-linear geophysical models. Comparative discussions of the performances and advantages of variational and ensemble DA methods can be found in, e.g. Kalnay et al. (2007), Bannister (2017) and Carrassi et al. (2018). As they aim at solving similar problems, a recent tendency is to combine both methods to benefit from their respective advantages.*

– p. 13, l. 4-13: There is a now long range of DA literature on how to estimate model bias. One good reference is the following:

Dee, D. P. Bias and data assimilation, Q. J. Roy. Meteor. Soc., 131, 3323–3343, doi: 10.1256/qj.05.137, 2005.

Could you reflect on that possibility in your discussion?

– p. 13, l. 14-20: Same comment as before on estimating observation error covariances matrices. A good review paper:

Tandeo, P., Ailliot, P., Bocquet, M., Carrassi, A., Miyoshi, T., Pulido, M. and Zhen, Y.: Joint Estimation of Model and Observation Error Covariance Matri- ces in Data Assimilation: a Review. Mon. Weather Rev., submitted, available at: https://arxiv.org/abs/1807.11221v2, 2018.

Most approaches are based on Desroziers diagnostics, see:

Desroziers, G., Berre, L., Chapnik, B. and Poli, P.: Diagnosis of observation, background and analysis-error statistics in observation space. Q. J. Roy. Meteor. Soc., 131, 3385–3396, doi: 10.1256/qj.05.108, 2005.

For the two points above, I have slightly reformulated the discussion and added the references :
*In a review paper Tandeo et al. (2018) illustrate the impacts of badly calibrated observation and model error covariance matrices in a sequential DA framework and discuss available methods and challenges for their joint estimation. For the question of the impact of systematic errors, i.e. bias, either in the model and in the observations, and their correction by augmenting the system state in variational and ensemble DA, interested readers are referred to Dee (2005).*

**MINOR COMMENTS AND TYPOS**

– p. 1, l. 5: "starting FROM this initial state . . ."

Done

– p. 3, l. 15: "the Kalman filter analysis is REWRITTEN and . . ."

Done

– p. 3, l. 16: The references you mention are all about deterministic versions of the EnKF (Pham et al., 1998, SEIK filter; Bishop et al., 2001, ETKF filter: Nerger et al., 2012, ESTKF filter). But not every EnKF has a deterministic analysis, the stochastic EnKF has also been an important part of EnKF history. Could you add the following references to make your point broader?

Burgers, G., van Leeuwen, P. J., and Evensen, G. Analysis scheme in the ensemble Kalman filter, Mon. Weather Rev., 126, 1719–1724, doi: 10.1175/1520- 0493(1998)126<1719:ASITEK>2.0.CO;2, 1998.

Houtekamer, P. L., and Mitchell, H. L. Data assimilation using an ensemble Kalman filter technique, Mon. Weather Rev., 126, 796-811, doi: 10.1175/1520- 0493(1998)126<0796%3ADAUAEK>2.0.CO%3B2, 1998.

References added. I do not introduce in the paper this distinction between stochastic and deterministic filters, as I think it would add more confusion for the reader. As suggested in the sentence just after, interested readers may refer to the review paper by Vetra-Carvalho et al. (2018).

– p. 3, l. 23: "However the many applications in meteorology and oceanography show . . ." While EnKFs have been primarily developed for those two applications, it has been successfully used in a wide range of applications, from hydrology, to crop modelling and oil extraction. Could you rephrase the sentence to show the broad range of applications for EnKF including some that may be closer to glaciology? Maybe add other references too?

this has been changed to:
*However, the many applications in geoscience with large and non-linear models have shown that the method remains robust in general and EnKFs are used in several operational centres with atmosphere, ocean and hydrology models (e.g. Sakov et al., 2012; Houtekamer et al., 2009; Hendricks Franssen et al., 2011). While firstly developed for numerical weather and ocean prediction where the forecasts are very sensitive to the model initial state, the method is also widely used, e.g. in hydrology, for join state and parameters estimations (Sun et al., 2014).*

– p. 6, Eq. (13): Could you define $\bar{X}^f$ ?

Done

- p. 9, l. 13: "the transient ASSIMILATION ON model projections"

  Done

---

## Referee Report (RR1)

**Review of "Assimilation of surface observsations in a transient marine ice sheet model using an ensemble Kalman filter", Fabian Gillet-Chaulet**

This paper presents a methodology for improving the initialisation of basal conditions from observed surface data for ice sheet models using a time-dependent initialisation. The author uses an established method – ensemble Kalman filter – to assimilate transient observations of surface elevation and velocities into a marine ice sheet model with a moving grounding line. The method is tested on an idealised marine ice sheet that is in the early stages of an unstable retreat using a twin experiment. Previous use of this method was for a flowline shallow ice method (Bonan et al., 2014).

I can see that the paper has been through a thorough technical review in the previous round, and that the author dealt with the comments thoroughly and in detail. New figures have been added that aid the understanding of the method and to further explore the results. I particularly like Figure 1, which gives a clear outline of how the method works and provides a comparison with the maybe more well-known 4D time-dependent variational methods. This figure really adds to the manuscript and makes it more accessible. I do agree with the author that the paper now constitutes a clear introduction to the use of these methods in glaciology. The technical aspect of the paper is well explained and clear as a result of the review process already undertaken.

The results from the idealised twin experiment look promising, with good agreement between a reference forecast and data-assimilated forecast when the assimilation period is sufficiently long (20-35 years). The results are well presented and well explained. The paper is detailed, well written and gives a thorough discussion of the strengths and limitations of the method, in particular detailing the main challenges that need to be overcome before generalising this method for use in large-scale ice-sheet modelling, which will be of interest to many other scientists.

Overall, I find it to be an interesting and worthwhile addition to the literature in this area, and I expect it will be of use to a range of other scientists working towards using such algorithms in realistic simulations. In view of this, and the other reviews and subsequent response from the author, I recommend publishing this article with only a few minor revisions.

Minor points:

1. From looking at the results for 20 years versus 35 years, it seems that we need quite a long observation record in order to use this method successfully -i.e. in Figure 12 and page 16, second paragraph – 20 years doesn't seem like long enough to get the benefits from using this method. Can you comment on that? Would we expect to have such data sets available soon?
2. Page 11, Line 30: So volume isn't conserved, but you say that estimation of ice sheet volume should improve as more data are assimilated. So you should converge on the true volume using this method? Can you directly compare the reference volume with results at the end of the DA process? Actually, Figure 12 shows convergence of VAF change at t=100 – should this figure be referenced here?
3. Page 11, Line 33 – clarify this sentence please: "as smoother might be more appropriate and the smoother extension of the ESTKF  can be found in…". Do you mean "a smoother", and what about a 4D-var method?
4. Page 17, L5: I think this is the first time grid size is stated? If so, I think it'd be helpful to state it earlier.

5. Page 17: I'm confused about the "effective observation dimension" – can you explain why it is 56?
6. Figure 5: green dots are not clearly visible in the top panel.
7. Table A1: Zs is listed as both top surface elevation and bottom surface elevation
8. "informations" appears a few times in the text rather than "information"

---

## Author Response (AR2)

Dear Editor, dear reviewers,

Many thanks for the very positive comments on the revised version of the manuscript. You will find below my point by point response to the remaining comments.

**Anonymous Referee #2**

I am very pleased with the updated version of the manuscripts and I suggest to accept the paper subject to technical corrections that I list below:

p. 8, l. 4: replace "this is why there is several EnKFs that exactly satisfy …" by "this is why several EnKFs exactly satisfy …"

Done

p. 8, l. 17: "EnKFs as Monte-Carlo methods can suffer from …". Remove the "can", EnKFs for large-scale problems always suffer from undersampling issues.

Done

p. 8, l. 18: "the analysis adjustS ..."

Done

p. 8, l. 32: "they should yield to similar results". Remove "should".

Done

p. 9, l. 9: Replace "this shouldn't be" by "this should not be"

Done

p. 15, l. 1: "Similar conclusions are drawn IF the assimilation is pursued"

Done

p. 15, l. 2: "RMSEu SHOW a higher variability"

Done

p. 15, l. 3: "RMSEb" instead of "RMEb"

Done

p. 15, l. 11: "However, it DOES NOT necessarily"

Done

**Anonymous Referee #3**

This paper presents a methodology for improving the initialisation of basal conditions from observed surface data for ice sheet models using a time-dependent initialisation. The author uses an established method – ensemble Kalman filter – to assimilate transient observations of surface elevation and velocities into a marine ice sheet model with a moving grounding line. The method is tested on an idealised marine ice sheet that is in the early stages of an unstable retreat using a twin experiment. Previous use of this method was for a flowline shallow ice method (Bonan et al., 2014).

I can see that the paper has been through a thorough technical review in the previous round, and that the author dealt with the comments thoroughly and in detail. New figures have been added that aid the understanding of the method and to further explore the results. I particularly like Figure 1, which gives a clear outline of how the method works and provides a comparison with the maybe more well-known 4D time-dependent variational methods. This figure really adds to the manuscript and makes it more accessible. I do agree with the author that the paper now constitutes a clear introduction to the use of these methods in glaciology. The technical aspect of the paper is well explained and clear as a result of the review process already undertaken.

The results from the idealised twin experiment look promising, with good agreement between a reference forecast and data-assimilated forecast when the assimilation period is sufficiently long (20-35 years). The results are well presented and well explained. The paper is detailed, well written and gives a thorough discussion of the strengths and limitations of the method, in particular detailing the main challenges that need to be overcome before generalising this method for use in large-scale ice-sheet modelling, which will be of interest to many other scientists.

Overall, I find it to be an interesting and worthwhile addition to the literature in this area, and I expect it will be of use to a range of other scientists working towards using such algorithms in realistic simulations. In view of this, and the other reviews and subsequent response from the author, I recommend publishing this article with only a few minor revisions.

Minor points:

1. From looking at the results for 20 years versus 35 years, it seems that we need quite a long observation record in order to use this method successfully -i.e. in Figure 12 and page 16, second paragraph – 20 years doesn't seem like long enough to get the benefits from using this method. Can you comment on that? Would we expect to have such data sets available soon?

   As shown in Fig. 3 and Fig. 9 each assimilation improves the reconstruction of the basal conditions and the model surface velocities and elevation. And after 20 years of assimilation, rates of retreat predicted by the model are much more accurate. However, what remains uncertain is the date at which the instability will occur. If the assimilation is pursued up to this date, all the members are pushed in the instability and the spread of the forecast around the truth is largely decreased. So I think the main difference is not 20 vs 35 years, but instability shown or not in the observations.

   I Have made slight changes in the discussion page 16:

« In addition, the ensemble framework naturally allows to estimate and propagate the uncertainty of the estimated parameters. *Each assimilation of new data improves the reconstruction of the basal conditions (Fig. 3), and the first ten years are very efficient in reducing error and the spread of the model surface velocities and elevation (Fig. 9). Furthermore, we have shown that the remaining uncertainties in the basal conditions do not significantly affect GL retreat rates once the unstable retreat is engaged. However, they can lead to considerable delays in the initiation of the instability. If the assimilation is pursued up to the beginning of the instability (35 years in our experiment) all the members exhibit the unstable retreat and centennial-scale model projections converge to the reference (Fig. 12 )* ».

And added one sentence in the conclusion:

*"However, if the assimilation is pursued up to a time when the glacier is engaged in the unstable retreat, all the members exhibit the instability and the spread of centenial-scale model projections, in terms of volume and grounding line position, is largely reduced."*

2. Page 11, Line 30: So volume isn't conserved, but you say that estimation of ice sheet volume should improve as more data are assimilated. So you should converge on the true volume using this method? Can you directly compare the reference volume with results at the end of the DA process? Actually, Figure 12 shows convergence of VAF change at t=100 – should this figure be referenced here?

Very interesting question, in fact it appears that we expect that the filter will improve the estimation of the state variables $zs$ and $b$, and thus will provide a better estimate of the ice thickness. So we expect that the filter will provide the best estimate of the volume distribution. Interestingly, by error compensation, we may expect that an *a priori* with a totally different thickness distribution, and thus high rms on $zs$ and $b$, could lead to a perfect estimation of the ice volume. So strictly speaking, there is no guaranty that the filter will improve the estimation of the total volume. However you are true, a better initial state improves volume projections as shown in Fig. 12.

I have changed the discussion page 11 as follow:
*"Because both $z_S$ and b are included in the state vector, the analysis does not conserve the ice sheet volume, neither for the ensemble mean and the individual members. However, as illustrated in Fig. 1, the estimation of $z_S$ and b, and thus of the ice thickness, should be improved at each analysis as more data are assimilated, and the final state is the best estimation provided by the filter knowing the model, all the observations during the assimilation window and their uncertainties. As mentioned in the introduction, if the main interest is an analysis of past volume changes, a smoother or a variational method might be more appropriate. The smoother extension of the ESTKF can be found in Nerger et al. (2014). Note however that, interestingly, if we expect that the filter will improve the estimation of the ice thickness there is no guaranty in general that it will provide a better estimate of the total volume as an a priori with a totally different thickness distribution could lead, by compensation of the errors, to a perfect estimate of the true volume."*

As this section describe the assimilation set-up I do not make reference to results in this paragraph.

Looking at the model results, the volume above flotation (VAF) for x>300km, is underestimated by only 0.41% by the a priori (taking the noisy observed surface at t=35a and the mean of the

initial bed elevations (shown in Fig. 5)), it is underestimated by 0.24% by the mean of the assimilation at t=35a. As I already show and discuss many results, I prefer not to add a paragraph to discuss this result.

3. Page 11, Line 33 – clarify this sentence please: "as smoother might be more appropriate and the smoother extension of the ESTKF can be found in…". Do you mean "a smoother", and what about a 4D-var method?

    Yes you are right, this has been changed to : *« As mentioned in the introduction, if the main interest is an analysis of past volume changes, a smoother or a variational method might be more appropriate. The smoother extension of the ESTKF can be found in Nerger et al. (2014). »*

4. Page 17, L5: I think this is the first time grid size is stated? If so, I think it'd be helpful to state it earlier.

    No the mesh size is defined page 9 in Sec. 3.1 Reference simulation , and given in Table A1.

5. Page 17: I'm confused about the "effective observation dimension" – can you explain why it is 56?

    This part now reads : *« Using three different models, Kirchgessner et al. (2014) have shown that good performances are obtained when r is such that the effective local observation dimension, defined as the sum of the weights attributed to each observation during the local assimilation, is equal to the ensemble size. **Here the observation weights decrease with the distance to the local assimilation domain following a fifth-order polynomial function mimicking a Gaussian function (Gaspari and Cohn, 1999)**. The value r = 8km used for the 50 members-ensemble gives an effective observation dimension of 56. »*

6. Figure 5: green dots are not clearly visible in the top panel.

    It shows bigger triangles now

7. Table A1: Zs is listed as both top surface elevation and bottom surface elevation

    Corrected

8. "informations" appears a few times in the text rather than "information"

    Done

[revised manuscript text omitted]